# Scaling Laws for Differentially Private Language Models

Ryan McKenna [1]   Yangsibo Huang [1]   Amer Sinha [1]   Borja Balle [2]   Zachary Charles [1]
Christopher A. Choquette-Choo [2]   Badih Ghazi [1]   Georgios Kaissis [2]   Ravi Kumar [1]   Ruibo Liu [2]   Da Yu [1]
Chiyuan Zhang [1]

## Abstract

Scaling laws have emerged as important components of large language model (LLM) training as they can predict performance gains through scale, and provide guidance on important hyperparameter choices that would otherwise be expensive. LLMs also rely on large, high-quality training datasets, like those sourced from (sometimes sensitive) user data. Training models on this sensitive user data requires careful privacy protections like differential privacy (DP). However, the dynamics of DP training are significantly different, and consequently their scaling laws are not yet fully understood. In this work, we establish scaling laws that accurately model the intricacies of DP LLM training, providing a complete picture of the compute-privacy-utility trade-offs and the optimal training configurations in many settings.

## 1. Introduction

Large language models (LLMs) are revolutionizing how we interact with technology, powering everything from instant translations and concise summaries to complex reasoning and creative content generation (Achiam et al., 2023; Gemini Team, 2023). Training increasingly large models on ever larger datasets is a key success factor for these LLMs, with frontier models being trained for millions of GPU-hours (Dubey et al., 2024) and trillions of tokens (Abdin et al., 2024; Gemma Team et al., 2024a;b). Scaling laws for neural language models are crucial because they provide a framework for understanding and predicting the performance gains achievable with increased computational resources, and importantly, guide the optimal allocation of that compute budget between model size and dataset size (Kaplan et al., 2020; Hoffmann et al., 2022).

[1]Google Research [2]Google DeepMind. Correspondence to: Ryan McKenna <mckennar@google.com>.

*Proceedings of the 42$^{nd}$ International Conference on Machine Learning*, Vancouver, Canada. PMLR 267, 2025. Copyright 2025 by the author(s).

The scale of data driving LLM progress also creates a critical privacy challenge. State-of-the-art models train on massive, diverse datasets (Dubey et al., 2024; Gemma Team et al., 2024a) that are also distributed (Carlini et al., 2024) making it difficult to exclude inadvertently shared personal information. Paradoxically, user data, a key privacy concern, is also crucial for advancing LLM capabilities. User interactions provide invaluable feedback for generating realistic synthetic data (Xie et al., 2024; Kurakin et al., 2023) and aligning models with human values (Stiennon et al., 2020), reflecting real-world use cases better than web-scraped text. However, direct training on sensitive user data is risky due to memorization and regurgitation (Carlini et al., 2021; 2023; Ippolito et al., 2023; Lukas et al., 2023; Biderman et al., 2023; Prashanth et al., 2025). This tension—the need for user data versus protecting user privacy—is addressed by differential privacy (DP) (Dwork et al., 2006).

While DP offers a principled solution to the tension between data utility and privacy in LLM training, applying it in practice, especially to large-scale models, presents significant challenges. DP mechanisms like DP-SGD (Abadi et al., 2016) and its variants introduce computational overhead, implementation complexity (Subramani et al., 2021), and utility degradation (Bassily et al., 2014). While it is well-known that DP-SGD benefits substantially from training with very large batch sizes (Anil et al., 2022; De et al., 2022; Ponomareva et al., 2023), little work has been done to understand the conditions under which this holds in compute-constrained settings, i.e., when an increase in batch size must be coupled with a decrease in model size or number of iterations. In part due to this reliance on large batch sizes, the largest models trained with DP today have hundreds of millions, rather than billions, of parameters (Anil et al., 2022; Li et al., 2022; Berrada et al., 2023; Ghalebikesabi et al., 2023; Charles et al., 2024; Sander et al., 2023).

To train large models with DP, it is crucial to spend both the compute budget *and* the privacy budget judiciously. In this work, we pave the way towards training at the billion-parameter scale by initiating a study on the *scaling laws of DP training*. To that end, we extend traditional scaling laws to consider a compute-privacy-utility trade-off, accounting for intricacies and additional variables introduced by DP

training. Through a rigorous set of experiments, we empirically model this trade-off, and provide a thorough analysis of these experimental results to answer a number of scaling law-style questions, finding (among other things) that:

- The compute budget allocation predicted by non-private scaling laws is far from optimal under DP, even for huge privacy budgets, confirming the need for our study.
- However, we can accurately predict the optimal breakdown of the compute budget into model size, batch size, and iterations for virtually any privacy budget and dataset size. These compute-efficient training configurations save $5\times$ to $100\times$ compute compared to baseline configurations, while retaining comparable privacy and utility.
- The optimal model size is typically at least an order of magnitude smaller with DP than without. This provides insight into the challenges of training large billion-parameter or larger language models with DP.
- In the DP setting, increasing the compute budget can sometimes yield little to no reduction in the loss unless accompanied by a corresponding increase in the privacy budget or dataset size.

## 2. Preliminaries and Problem Setup

Our dataset $\mathcal{D}$ consists of text sequences, where each individual contributes a single sequence $\mathbf{x} = (x_1, \ldots, x_S)$ of $S$ tokens, and each token is drawn from a predefined vocabulary $\mathcal{V}$. We let $N$ denote the total number of individuals contributing to the dataset.

| Key | Definition |
|-----|------------|
| $\epsilon$ | Privacy budget |
| $N$ | Data budget |
| C | Compute budget |
| $B$ | Batch size |
| $T$ | Iterations |
| $S$ | Sequence length |
| $M$ | Model parameters |
| $\bar{\sigma}$ | Noise-batch ratio |

**Masked Language Modeling.** In this work we focus on the masked language modeling task (Devlin et al., 2019), where each sequence has a chosen fraction $p_{\text{mask}}$ of tokens masked out, i.e., replaced with a special masking token [MASK], uniformly at random. The goal is to predict the original token for each masked token using the entire context (bidirectionally). Let $\bar{\mathbf{x}}$ represent the original sequence of tokens but masked using the above procedure and $\mathcal{M}$ the ids of the masked tokens in $\bar{\mathbf{x}}$. For a given parameter vector $\theta \in \mathbb{R}^M$, the language model defines a conditional probability $p_\theta(x_j \mid \bar{\mathbf{x}})$ for each $j \in \mathcal{M}$, and the goal is to find $\theta$ to maximize the likelihood of all masked training tokens.

**Differential Privacy.** A randomized mechanism $\mathcal{A}$ satisfies $(\epsilon, \delta)$-*DP* (Dwork et al., 2006) if, for any two datasets $\mathcal{D}$, $\mathcal{D}'$ that differ by a single individual, all subsets $\mathcal{O}$ of possible outputs of $\mathcal{A}$ and $\epsilon > 0, 0 \leq \delta < 1$:

$$\Pr[\mathcal{A}(\mathcal{D}) \in \mathcal{O}] \leq e^\epsilon \Pr[\mathcal{A}(\mathcal{D}') \in \mathcal{O}] + \delta.$$

---

**Algorithm 1** (Informal) Generalized DP-SGD. Appendix B.1 discusses the informalities.

---

**Input:** Dataset $\mathcal{D}$, noise-batch ratio $\bar{\sigma}$, (expected) batch size $B$, iterations $T$
**Output:** Model parameters $\theta$.
Initialize model parameters $\theta_0 \in \mathbb{R}^M$
  **for** $t = 1$ **to** $T$ **do**
    Select a (possibly random) size $\approx B$ minibatch $\mathcal{B}_t \subset \mathcal{D}$
    $\bar{g} = \frac{1}{B} \sum_{\mathbf{x} \in \mathcal{B}_t} \text{clip}(\nabla \ell(\theta_{t-1}; \mathbf{x}))$
    $\tilde{g} = \bar{g} + \bar{\sigma} \mathcal{N}(0, 1)^M$
    $\theta_t = \text{OptimizerUpdate}(\theta_{t-1}, \tilde{g})$
**return** $\theta_T$

---

**DP-SGD.** DP-SGD is a widely used algorithm to train neural networks with DP. It attains provable DP guarantees through limiting the contribution (*sensitivity*) of each example by clipping its gradient to some $\ell_2$-norm (wlog, 1), and then adding isotropic Gaussian noise to the averaged clipped gradients; see Algorithm 1 for pseudo-code. Our algorithm is a slight generalization of the original DP-SGD (Abadi et al., 2016): to enable adaptive optimizers, which are often crucial for training transformer models, the subroutine OptimizerUpdate can be any first-order optimizer. Throughout this work, we set OptimizerUpdate to be Adam (Kingma & Ba, 2015), which we denote DP-Adam. Algorithm 1 satisfies a formal DP guarantee that can readily be computed as a function of $\bar{\sigma}$, $B$, $N$, and $T$ using a suitable privacy accountant. The dp_accounting library provides functions that can efficiently and tightly compute the minimum value of $\bar{\sigma}$ as a function of $\epsilon$, $\delta$, $N$, and $B$ (Google DP Team, 2022).

**Noise-Batch Ratio.** Note that we parameterize Algorithm 1 in terms of the *noise-batch ratio* $\bar{\sigma}$, which is the standard deviation of noise added to the mean minibatch gradient, instead of the usual noise multiplier which is typically added to the summed minibatch gradient. While the noise multiplier typically governs the privacy properties of the mechanism, the noise-batch ratio is a better proxy for the downstream learning performance. Specifically, there are two sources of variance in the stochastic gradient estimate $\tilde{g}$: (1) the minibatch estimate of the true population gradient and (2) the Gaussian noise added to ensure DP. Prior work has shown that the latter dominates the variance in most practical regimes (Ponomareva et al., 2023).

### 2.1. Compute-Optimal DP Training

We are interested in empirically modeling how the compute-privacy-utility trade-off changes as a function of the problem parameters. We follow ideas used to model the compute-utility trade-off in the non-private setting (Kaplan et al., 2020; Hoffmann et al., 2022), but extend them to study the private setting by additionally considering the *privacy budget* and *data budget*. The key concepts are:

- **Compute Budget** ($C$) refers to the total floating point

operations (FLOPs) required to train the model. We use the standard approximation of Kaplan et al. (2020): $6 \cdot M \cdot B \cdot S \cdot T$ to measure this, which is proportional to the model size ($M$) and the total number of training tokens ($B \cdot S \cdot T$). Note that unlike the non-private scaling laws, we use $B$ to represent the number of examples in a batch (not tokens) because this quantity is what matters for privacy calculations. This approximation provides a platform-independent estimate of compute requirements, and is justified further in Appendix B.3.

- **Privacy Budget** ($\epsilon$) refers to the value of $\epsilon$ at fixed $\delta$ in ($\epsilon,\delta$)-DP. We fix $\delta = 10^{-8} = \Theta(1/N)$ unless otherwise mentioned, which is a common choice in the literature (Abadi et al., 2016).
- **Data Budget** ($N$) refers to the number of individuals in the training dataset, $|\mathcal{D}|$, which can be different from the number of examples processed by DP-SGD under multiple passes. Note that our analysis and insights also hold in the more general setting where individuals can contribute multiple examples, although the data budget must still be interpreted as the number of individuals rather than the number of examples (see Appendix B.2).

The privacy and data budgets are absent in most non-private scaling laws because they often assume that an infinite stream of data is available and no privacy protections are needed. In the private setting, model training is often constrained by both a fixed data budget (i.e., a limited set of examples) and a fixed privacy budget (i.e., $\epsilon$ in DP). Both of these impact model training; thus, it is crucial to determine the optimal compute usage given the constraints on privacy and data, by fitting a scaling law accounting for this.

### 2.2. Private Scaling Law Challenges

**Additional Scaling Factors.** As mentioned above, our private scaling laws account for the additional data and privacy considerations not present in the non-private scaling law studies. These add complexity because DP adds noise beyond what is introduced through the stochasticity of training. Without DP, training with a batch size of $B$ for $T$ iterations is roughly equivalent to training with a batch size of $1$ for $B \cdot T$ iterations, as long as $B$ is below the so-called "critical batch size" (McCandlish et al., 2018; Shallue et al., 2019; Zhang et al., 2025). However, this relationship does not hold in DP settings, and further, DP training requires larger batch sizes to mitigate the impact of the added noise (Anil et al., 2022; De et al., 2022).

**Compute Requirements.** Even without DP, exhaustive hyperparameter tuning is infeasible for large models. DP training introduces further complexity with additional hyperparameters and the need to adapt standard defaults (e.g., learning rate) to new regimes, necessitating careful protocol design to achieve near-optimal selection within reasonable

compute. Further, it is important to consider that collapsing the privacy and data budgets to a single quantity is unlikely to provide generalizable insights.

## 3. Private Scaling Law Methodology

In this section, we detail our methodology for estimating the validation cross-entropy loss from model size, noise-batch ratio, and training iterations, which in turn lets us estimate the utility under a fixed compute, privacy, and data budget.

### 3.1. Decoupling Noise Calibration

A key part of our methodology is to directly analyze the impact of the noise-batch ratio for a fixed but reasonably large *physical batch size*, rather than indirectly through changes to the privacy budget or batch size. Via *post-hoc* accounting, we will predict what could happen at different *hypothetical batch sizes*, an approach that is justified by the fact that typically the noise-batch ratio is the primary source of noise in the minibatch gradients, outweighing the noise due to minibatch sampling (Ponomareva et al., 2023).

This decoupling enables for a better understanding of the underlying trade-offs. Without this approach, the non-linearities in DP accounting (detailed in Section 4.5) make it difficult to assess these. We note that a naive methodology that tries to directly model the scaling law as a function of privacy budget (without going through the noise-batch ratio) would either provide less insight (by not generalizing across data budgets), or require much more compute.

After decoupling, the function we want to fit requires three inputs: the model size $M$, the number of iterations $T$, and the noise-batch ratio[1]. We require the function to be well-defined for a broad range of possible inputs that could be encountered in practical settings. We also need it to cover extreme points that may not be likely to be useful in practice, but may provide additional scientific insight. The methodology described below attempts to balance this need with the goal of using compute responsibly.

### 3.2. Detailed Experimental Setup

**Models and Datasets.** We train BERT models ranging in scale from Tiny (4M parameters) to Mega (778M parameters), summarized in Table 1. We focus on the default BERT dataset, which includes approximately 3.3B words (Zhu et al., 2015; Devlin et al., 2019) before tokenization. Each example is truncated or padded as necessary to a sequence of fixed length $S = 512$.[2]

---

[1]The learning rate is a hyperparameter that is optimized over and not modeled directly.

[2]Future work could fruitfully consider other sequence lengths, as they are likely to showcase interesting trade-offs.

*Table 1.* Models used in this study, taken from Devlin et al. (2019).

| Model | Layers | Heads | Dims | Params ($M$) |
|---|---|---|---|---|
| BertTiny | 2 | 2 | 128 | 4.5M |
| BertMini | 4 | 4 | 256 | 11.4M |
| BertSmall | 4 | 4 | 512 | 29M |
| BertMedium | 8 | 8 | 512 | 41M |
| BertBase | 12 | 12 | 768 | 109M |
| BertLarge | 24 | 16 | 1024 | 335M |
| BertMega | 24 | 24 | 1536 | 778M |

**Optimizer.** We use DP-Adam throughout. We use 1000 steps of learning rate warm-up, followed by exponential learning rate decay, decreasing the learning rate by a factor of $10\times$ over a horizon of 128K iterations. We use per-example clipping with an $\ell_2$ clip norm of 1.0 across all experiments. We employ the normalized variant of clipping proposed by De et al. (2022), to help decouple learning rate tuning from clip norm. We verified that this setting effectively clips most per-example gradients, as recommended in prior work (Li et al., 2022; De et al., 2022).

**Learning Rates.** We tune the learning rate with per-example gradient clipping but no noise, finding that the optimal learning rate is consistently $2^{-7}$ across all model scales. With noise, we consider three learning rates: $2^{-7}, 2^{-8}, 2^{-9}$. This methodological choice was based on early ablations that showed that when adding noise the optimal learning rate does decrease, but gradually so; see Appendix C.7.

**Batch Sizes.** We use a fixed physical batch size of 1024 across all experiments. Via *post-hoc* accounting, we will analyze what *could* happen at different hypothetical batch sizes, under the assumption that cross-entropy primarily depends on the privacy budget and batch size through the noise-batch ratio. We may expect this choice underestimates the benefit of larger batch sizes, a question we study empirically in Appendix C.3.

**Noise-Batch Ratio.** We consider 18 values of noise-batch ratio: $\{2^{-k} \mid k = 6, \ldots, 23\}$, plus a baseline value of 0 corresponding to non-private training.

**Metrics.** Every 100 training iterations, we record the average training loss over the previous 100 iterations (or $102,400$ training examples). Using training loss instead of evaluation loss is standard practice in scaling laws work, and is justified by the fact that we are training for less than a single physical epoch, so training loss is an unbiased estimate of evaluation loss.

We provide details on the compute platforms and training throughput in Appendix C.5.

### 3.3. Semi-Parametric Modeling

After training the models described above, we obtain a grid of measurements over 6 unique model sizes, 1280 unique number of iterations, 18 unique noise-batch ratios, and three learning rates. While one can directly query this data to answer a variety of interesting questions, we ultimately need to know what might happen in-between (and possibly outside of) the grid points we specifically evaluated. For that, we need to fit a function to the data, for which we follow a semi-parametric approach. See Appendix E for studies with fully parametric fits.

**Data Cleaning and Smoothing.** First, we note that loss *should* monotonically increase with increased noise-batch ratio, and monotonically decrease with increased iterations (unless training diverges), and we want our fitted function to capture this property. In practice, this invariant only holds approximately due to inherent variance in the training process. To clean the data, we apply the following post-processing steps:

1. For each model size and noise-batch ratio, we apply a rolling average over the 10 previous measurements to calculate a smoothed loss value. This corresponds to an average over $10 \cdot 100 \cdot 1024$ total examples, but does not perfectly preserve the expected invariant.
2. For each model size and noise-batch ratio we apply *isotonic regression* to ensure the 1280 loss values are monotonically decreasing with respect to the number of iterations. For each model size and number of iterations, we apply isotonic regression again to ensure the 18 loss values are monotonically increasing with respect to the noise-batch ratio. We do not enforce any monotonicity with respect to model size.

We use isotonic regression to enforce desired monotonicity properties, rather than simpler alternatives like taking the cumulative $\min$ across each dimension. The latter approach suffers from a statistical phenomenon known as the *minimum selection bias*, where one outlier sample can compromise the validity of the measurements. We visualize our smoothing process in Appendix C.9.

**Training Step Extrapolation.** Next, we extrapolate our smoothed data with respect to the number of iterations, by fitting a parametric form to the training curve and predicting where the loss would have gone if training continued beyond 128K iterations. We use a simple parametric form inspired by Hoffmann et al. (2022), namely $L = E + \frac{A}{T^\alpha}$. We fit this function using `scipy.optimize.curve_fit`, which uses the Levenberg–Marquardt algorithm to solve a nonlinear least squares problem (Nocedal & Wright, 1999). We independently fit a function for each model size and noise-batch ratio on data from iterations $16K$ to $128K$.

**Scaling Law Fitting.** After data cleaning, our goal is to fit a function $L(M, T, \bar{\sigma})$ that estimates the loss under a $M$-parameter model training for $T$ iter-

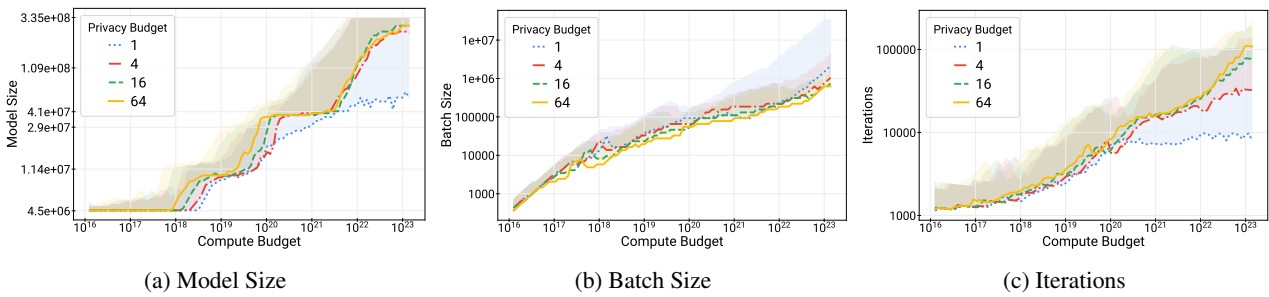

| (a) Model Size | (b) Batch Size | (c) Iterations |

*Figure 1.* Optimal model size, batch size, and iterations for varying privacy and compute budgets, with a fixed data budget of $10^8$. Lines show minimum values for each hyper-parameter that achieve within 1% of optimal cross-entropy for constant-compute training. Shaded regions indicate the full range of near-optimal settings.

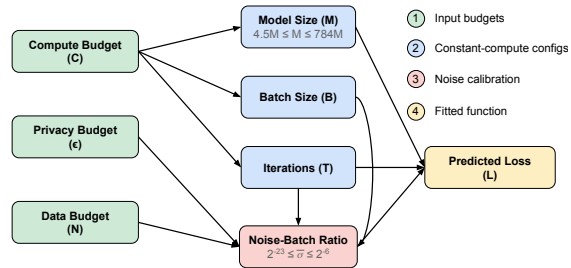

*Figure 2.* Workflow for estimating cross-entropy of different training configurations under given compute, privacy, and data budgets.

ations with a noise-batch ratio of $\bar{\sigma}$. We fit this function using linear interpolation, and specifically `scipy.interpolate.RegularGridInterpolator` in Python. Since $M$, $T$, and $\bar{\sigma}$ are all naturally varied in log-space, we apply interpolation to the function $F$ such that $F(\log M, \log T, \log \bar{\sigma}) := L(M, T, \bar{\sigma})$ instead. This function is well-defined for any $T$ and any $M, \bar{\sigma}$ within the range of experimental settings considered; that is, $M \in [4.5\text{M}, 784\text{M}], \bar{\sigma} \in [0.5^{23}, 0.5^6]$. Because we use interpolation, our fitted function matches the smoothed data exactly at the evaluation points, and approximates it in between them. In Appendix E we also fit a parametric form for this function as well, finding that it is largely consistent with the non-parametric fit.

### 3.4. Using the Fitted Functions

We are now able to answer DP scaling laws questions. Figure 2 summarizes our approach. We begin with inputs: the compute budget, privacy budget, and data budget. Second, we proceed by enumerating an exhaustive set of constant-compute training configurations; i.e., combinations of model size, batch size, and iterations that require the given compute budget. Using privacy accounting and noise calibration functions from the `dp_accounting` library, we compute the noise-batch ratio as a function of the privacy budget, data budget, iterations, and (expected) batch size. Finally, we query our fitted function with this noise-batch ratio, along with the given model size and number of iterations, giving

us a final estimate of the cross-entropy of these training configurations. In addition, we can also specify directly the training configurations instead of the compute budget for the purposes of conducting specific ablations or comparisons.

## 4. Experimental Findings of Scaling Laws

### 4.1. Optimal Compute Budget Allocation

We first determine how to best utilize our compute budget in different situations. Specifically, for a given compute/privacy/data budget, we aim to understand how to optimally allocate our compute budget among the model size, batch size, and number of iterations. Additionally, we seek to understand how the optimal allocation changes per budget. While this question can be answered for virtually any setting of the budgets with the data we collected, we visualize a few relevant slices of the data in Figure 1. More comprehensive results can be found in Appendix C.8. From this visualization, we make the following observations:

- For small compute budgets, the optimal allocation of compute budget does not exhibit a strong dependence on $\epsilon$. However, there is a small but consistent trend that with larger privacy budgets, one should train a larger model with a smaller batch size and for more iterations than one would train with a smaller privacy budget. This finding is somewhat surprising, since as the privacy budget gets larger, the point at which increasing batch size leads to diminishing returns in terms of noise-batch ratio increases roughly according to $\approx N\sqrt{\epsilon/T}$ (Ponomareva et al., 2023).

- There are many settings of model size, batch size, and number of iterations that achieve near-optimal loss, as indicated by the large shaded regions. This suggests some amount of robustness for compute-optimal training hyper-parameters. All else being equal, training smaller models on more tokens should generally be preferred due to their inference-time efficiency advantages.

- Optimal model sizes are much smaller than predicted by non-private scaling laws. For instance, at $10^{22}$ FLOPs,

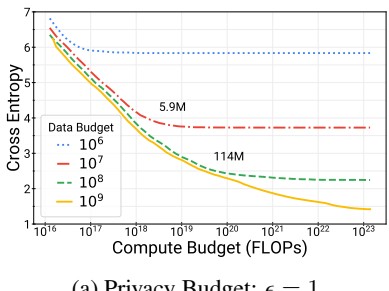
(a) Privacy Budget: $\epsilon = 1$

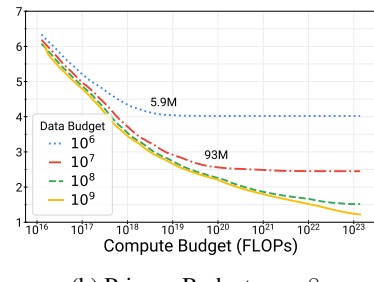
(b) Privacy Budget: $\epsilon = 8$

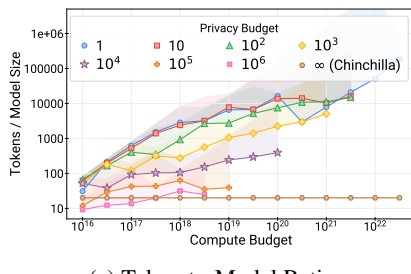
(c) Token-to-Model Ratio

*Figure 3.* (a-b) Best cross-entropy loss achieved for varying compute budgets, four data budgets, and two different privacy budgets. Each figure is annotated with the optimal model size at the inflection point for two of the curves. (c) Number of training tokens $S \cdot B \cdot T$ divided by number of model parameters for the compute-optimal training configuration, fixing the data budget to $N = 10^7$.

$\sim 10^8$ parameters are compute-optimal, compared to $\sim 10^{10}$ non-privately.

## 4.2. Benefits of Increased Compute

We now aim to understand and measure how much benefit increased compute budgets can provide and under *what* circumstances. In Figure 3a, we look at how the optimal achievable cross-entropy depends on the compute budget for different settings of data/privacy budget. Our main observations are:

- Increasing the compute budget can be a very effective strategy for reducing cross-entropy under a fixed privacy/data budget up to a limit, but there is an inflection point where increasing the compute budget beyond this point provides little to no benefit. The "critical compute budget" where this inflection point occurs increases with both privacy budget and data budget. For example, with a data budget of $10^8$ and a privacy budget of 1, the best cross-entropy is achieved with a compute budget $\gtrsim 10^{20}$ and corresponds to a model with 114M parameters. This is a qualitatively different behavior than non-private scaling laws, where increasing the compute budget continues to provide benefits even at the extreme scales.

More comprehensive analysis of the saturating compute budget for a representative set of data and privacy budgets can be found in Appendix C.1.

## 4.3. Token-to-Model Ratio

We now aim to understand more about compute-optimal training configurations, specifically the ratio of the number of training tokens (as measured by $S \cdot B \cdot T$) to model size and privacy budget. In other words, we study a form of sample complexity. In the absence of DP, a constant token-to-model ratio of $20\times$ is the recommended best practice (Hoffmann et al., 2022). As we see in Figure 3c, the behavior under DP is not as simple:

- The token-to-model ratio increases with compute budget, especially for smaller privacy budgets. As the privacy budget increases, the slope decreases, and for a sufficiently large privacy budget becomes nearly flat as predicted by the prior work. However, the privacy budget required to exhibit behavior similar to prior work is *extremely large*. Note that a privacy budget of $\epsilon = 1000$ provides no meaningful formal membership inference protection.[3] Nonetheless, the noise added still has a significant impact on training: its behavior in Figure 3c is more similar to a privacy budget of 1 than non-private training ($\epsilon = \infty$).
- For moderate privacy budgets in the range of $[1, 10]$, a good token-to-model ratio is typically between 1000 and 100, 000, although for sufficiently large compute budgets, it can go beyond this point. This connects back to an earlier observation that even with infinite compute, there is eventually no benefit to increasing the model size when using a modest privacy budget. These ratios roughly correspond to training models $10\times$ to $50\times$ smaller than predicted by Hoffmann et al. (2022).

## 4.4. Comparison Against Baselines

We now measure the improvement our compute-optimal training configurations provide over natural baselines. In the DP training literature, it is common to fix the training configuration (model, iterations, batch size), and vary the privacy budget. To that end, we consider 3 baseline training configurations: BertLarge trained for 7500 steps with a batch size of 1295, BertMedium trained for 5000 steps with a batch size of 15879 and BertTiny trained for 2500 steps with a batch size of 283, 061. In all three, we fix the data budget to $N = 10^7$. Each of these training configurations requires $10^{19}$ FLOPs. The first configuration is close to what would be predicted by non-private scaling laws (Hoffmann et al., 2022), while the last might be selected by an expert in DP

---

[3]However, values even larger than this have been shown to be effective against reconstruction attacks in prior works (Balle et al., 2022; Kaissis et al., 2023; Ziller et al., 2024).

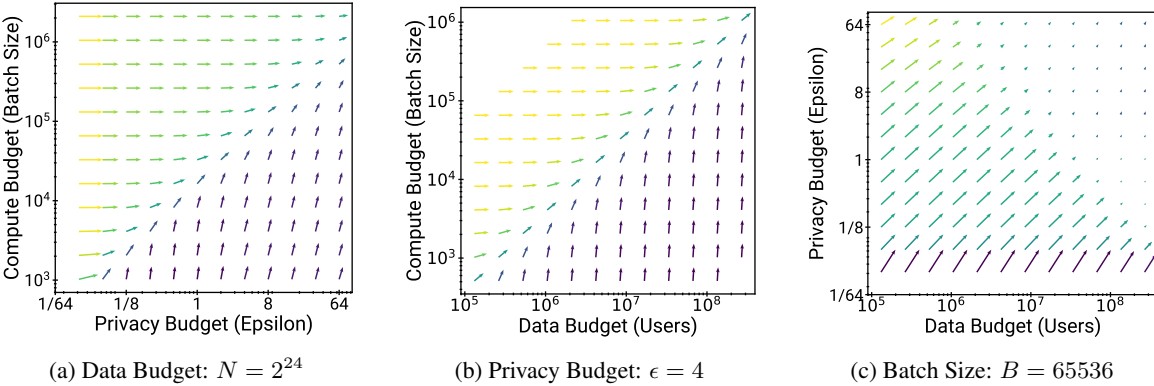

(a) Data Budget: $N = 2^{24}$      (b) Privacy Budget: $\epsilon = 4$      (c) Batch Size: $B = 65536$

*Figure 4.* Marginal benefits of increasing the privacy budget ($\epsilon$), compute budget ($B$), and data budget ($N$) on the noise-batch ratio.

who recognizes the importance of large batch sizes. The results are shown in Figure 5, from which we find:

- For most privacy budgets, the training configuration predicted by non-private scaling laws (BertLarge) yields very low utility. While utility improves for sufficiently large privacy budgets, this suggests that private scaling laws are fundamentally distinct from non-private ones.
- The optimal training configuration changes with the privacy budget, and naively using a fixed training configuration across all privacy budgets, as is common in the literature, leaves significant utility on the table.
- Compute-optimal training can either give better utility, or save compute/privacy budget under fixed utility. Training a compute-optimal model with $2 \times 10^{18}$ FLOPs yields similar utility as the best baseline models with $5 \times$ the FLOPs for the reasonable range of privacy budgets. This is just one instructive example. The savings in other settings may change depending on factors like data budget, compute budget, and quality of the baseline training configurations (e.g., the compute savings over BertLarge exceeds $100 \times$, although this is not shown).

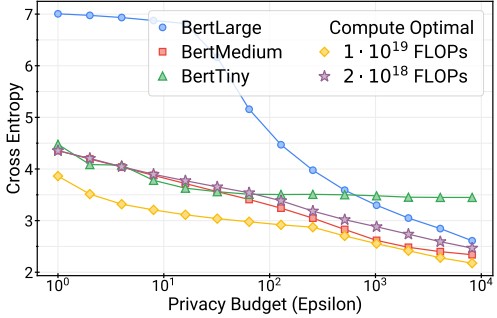

*Figure 5.* Comparison of a compute-optimal training configuration to some natural baselines as a function of the privacy budget. All models are trained with a compute budget of $10^{19}$ FLOPs and a data budget of $N = 10^7$ respectively.

### 4.5. Synergy between Privacy/Data/Compute Budgets

While many of the trade-offs that we explore in this work are data-dependent and require significant empirical investigation, many generalizable scaling insights can be derived purely by exploring privacy accounting. In this section we detail some of these, which corroborate many of our experimental evidence above and require very little compute. These insights are domain-agnostic, and therefore likely to generalize to other machine learning settings beyond language models, while also helping us understand and explain some of the experimental observations presented earlier.

We analyze how the noise-batch ratio behaves as a function of privacy budget (as measured by $\epsilon$), compute budget (as measured by $B$), and data budget (as measured by $N$). We fix $T = 16000$ training steps here, but our findings hold for any fixed number of steps[4]. We compute the noise-batch ratio for different settings by using the dp_accounting library (Google DP Team, 2022). Although the function that computes the noise-batch ratio is generally well-understood in the sense that we know how to compute it tightly given the privacy and training parameters, its precise behavior as a function of the privacy budget, compute budget, and data budget is not common knowledge. Indeed, due to lack of clear and simple guidance on how to configure DP-SGD, it is not uncommon to use or compare against sub-optimal configurations of DP-SGD.

In Figure 4 we plot three vector fields. Along each axis we vary the privacy budget, compute budget, and data budget. The direction and magnitude of the vectors indicate how much doubling each of these budgets reduces the noise-batch ratio. Each budget is varied on a logarithmic scale at different powers of 2. The length of the $x$ and $y$ components of the vector is determined by ratio of noise-batch ratio mi-

---

[4]While compute budget could also be varied through $T$, the effect of changing $T$ is data-dependent and the noise batch ratio is not directly comparable across different $T$.

nus one. For example, a vector of length 1 along the privacy budget axis means doubling the privacy budget reduces the noise-batch ratio by a factor of two.

As there are three budgets that together determine the noise-batch ratio and they interact in nuanced ways, we show three plots in Figure 4, where we vary two of the budgets at a time while fixing the third. These plots together provide a fairly complete picture of the behavior of the noise-batch ratio. Our main observations are enumerated below:

- In Figure 4a we see that varying the privacy budget or compute budget alone (while fixing the other) leads to diminishing returns. Increasing the privacy and compute budgets in tandem leads to consistent and predictable reductions in the noise-batch ratio.
- In Figure 4b we see a similar trend when varying data and compute budgets. At small compute budgets, increasing the data budget provides limited benefit, and vice-versa. Increasing them simultaneously leads to consistent and predictable improvements in the noise-batch ratio.
- In Figure 4c we see that while increasing data and privacy budgets can be helpful, for a fixed compute budget, increasing either provides diminishing and eventually negligible benefits.

These observations provide guidance on how to effectively configure DP-SGD and corroborate our scaling laws above.

## 5. Related Work

**Scaling Laws of Language Models.** Recent research has explored the scaling laws governing the performance of language models as they increase in size. Kaplan et al. (2020) found a power-law relationship between model size, dataset size, and compute budget, with performance on downstream tasks following predictable scaling curves. Hoffmann et al. (2022) extended this to open-ended language models, observing smooth scaling over 7 orders of magnitude. Chowdhery et al. (2022) trained PaLM, a 540B parameter model that continued the trends. These results suggest language models may continue improving as they scale, although Ganguli et al. (2022) note scaling alone may not be sufficient for open-ended intelligence. In the context of training language models with DP, where gradient clipping and noise addition (Abadi et al., 2016) alter training dynamics, the scaling laws have remained largely unexplored until this work.

**Applying DP in Fine-tuning or Prompting.** Recent studies demonstrate that fine-tuning (Bu et al., 2023; Wang et al., 2024; Du et al., 2023; Thaker et al., 2023; Zhang et al., 2024b; Tobaben et al., 2023; Wu et al., 2024a; Zhang et al., 2024a; Chua et al., 2024a) or prompting (Duan et al., 2023b;a; Wu et al., 2024b; Tang et al., 2024; Hong et al., 2024; Amin et al., 2024) LLMs can achieve strong performance while ensuring downstream data privacy. However,

these privacy guarantees are limited to downstream data, leaving the pre-training process exposed. Given that LLMs are pre-trained on extensive Internet data, which is often sourced without explicit user consent (Gold & Latonero, 2017), this raises ethical and privacy concerns (Tramèr et al., 2024). Safeguarding privacy during pre-training remains a significant challenge. This study seeks to provide new insights to advance privacy-preserving pre-training of language models.

**DP Training of Vision Models.** Training DP models from scratch for vision tasks is an active area of research (Yu et al., 2021; De et al., 2022; Bu et al., 2022; Kurakin et al., 2022; Sander et al., 2024). The most related work is that of Sander et al. (2023), who investigate the scaling behavior of DP training on vision tasks by varying key hyperparameters. They demonstrate that, under a fixed privacy budget, carefully tuning batch size, training steps, and learning rate is critical for better accuracy. However, they do not account for a bounded compute budget, a crucial factor in scaling law studies for language models (Hoffmann et al., 2022). Additionally, it remains unclear how their findings translate to language modeling tasks. In this work, we extend scaling law analyses to language models, incorporating both standard optimization hyperparameters and a bounded compute budgets to align more closely with recent LLM scaling research.

## 6. Conclusion and Future Directions

This work establishes a principled methodology for understanding the compute-privacy-utility trade-off of language models trained under DP, and it represents an important step towards training larger, more capable models efficiently on sensitive user data. This endeavor will require collecting increasingly larger datasets over larger groups of individuals, while simultaneously scaling up compute. For example, to train a billion parameter model optimally with DP, one could collect data from one billion individuals, using a generous privacy budget of $\epsilon \approx 10$, and train on large compute clusters for $\approx 10^{23}$ FLOPs. This is in stark contrast to non-private laws, e.g., Anil et al. (2023) suggest a much larger $\approx$ 20B parameter model could be trained with $\approx$ 2B examples.

This work raises several new questions worth exploring in future work, including how do the scaling laws change when (1) doing fine-tuning instead of pre-training, (2) using better underlying mechanisms, and (3) when allowed to vary the sequence length. These questions (along with several others) are discussed in greater detail in Appendix A.

## Impact Statement

This paper presents work whose goal is to advance the field of machine learning, specifically in the area of differentially private (DP) language models. It establishes DP scaling laws that shed light on the trade-offs between compute, privacy, and utility, and can lead to more efficient and effective methods for training LLMs on user data while satisfying DP, a gold standard for bounding the privacy loss. The scaling laws presented can help researchers and practitioners choose model sizes, batch sizes, and training iterations based on available compute, data, and privacy budgets. By developing methods to make DP training more feasible, the paper contributes to the responsible development and deployment of AI technologies. We point out that, when applying DP in practice, the privacy unit has to be chosen carefully; in particular, a user-level guarantee may be needed. Moreover, while a valuable tool, DP may not be sufficient when training on user data; additional mitigations may need to be simultaneously applied depending on the application.

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

# A. Limitations and Open Questions

While our methodology revealed a number of interesting findings about the behavior of scaling laws under DP, there are some limitations of our approach and questions that remain unanswered that we enumerate below.

**Fixed Physical Batch Size.**   Our methodology relies crucially on the assumption that the Gaussian noise introduced to preserve privacy far outweighs the randomness introduced from minibatch sampling, and thus it would be sufficient vary the noise-batch ratio while keeping the physical batch size fixed to a large constant value of 1024. Appendix C.3 reveals that this assumption may not be fully true, and that the physical batch size has a more nuanced effect that we cannot fully explain.

**Robustness to Other Training Setups.**   Our methodology focuses on a single class of BERT models, with a fixed dataset and DP mechanism, which allowed us to do deeper experimentation on other relevant variables. Our general methodology holds for different models, datasets, and mechanisms, but the exact quantitative findings may differ under different training setups. As the field continues to make advancements on training transformers with DP, it would be interesting and informative to rerun our experiments with better base mechanisms.

**Pre-training vs. Fine-tuning.**   As an important first step, we focused on the pre-training regime in this work, where we start with a completely random model which we train from scratch. Fine-tuning a pre-trained model with DP is often a preferable approach in practice to get the best privacy/utility trade-offs (Yu et al., 2022; Li et al., 2022). There are a number of challenges to overcome to quantify the scaling laws in this regime, but it remains an interesting question for future work.

**Sequence Length.**   Our experiments focus on a fixed sequence length of 512 tokens, which was the default value in the experiment we branched. However, the sequence length is yet another important knob that can be tuned alongside the batch size, model size, and number of iterations in language modeling tasks. There are likely interesting trade-offs to explore here: with smaller sequence lengths, less context is available to predict the next / missing tokens, but the saved computation can be used to increase the batch size, model size, or number of iterations. Whether the trade-off is worth it likely depends on the exact setting as well as the distributional properties of the training data.

**Over-Training and Inference-Time Compute.**   While this work focuses on the FLOPs required to pre-train a model to a given loss threshold, in practice language models are often over-trained in order to account for inference-time costs (Gadre et al., 2025). If a model is going to be deployed, it may make sense to over-train a smaller model (which is cheaper to serve) than to train a larger model for a compute-optimal FLOPs budget. While we do not study over-training in our work, we note that such a study is particularly fruitful in the case of DP training; the privacy costs already often favor smaller models (when compared to non-private scaling laws). Investigating this confluence would likely yield valuable insights into DP scaling laws.

**Larger Model Sizes.**   The accuracy of any given scaling law is predicated to some degree on the range of model sizes trained on. For example, Hoffmann et al. (2022) train model of up to 16B parameters. Due to the necessity of using very large batch sizes, training models of such scale requires a significant amount of compute. We leave the task of training on model of larger scale to future work, along with analysis of how much this affects the derived scaling law.

**Efficient Implementations of Per-Example Gradient Clipping.**   When considering to use a significant compute budget to train a large language model with DP, it is important that that model training code is carefully optimized to minimize the overheads of DP training. Using efficient vectorized per-example clipping implementations in JAX have been shown to work perform well with a reasonable overhead compared to non-private training (Subramani et al., 2021), although this focused on single-machine training scenarios, and more careful study is needed in this area when doing multi-machine training, especially when moving beyond pure data-parallelism, which we focused on in this paper.

**The Choice of Optimizer**   Our analysis relies on current optimization techniques, which may not be optimal for privacy-preserving training. Several potential optimizer improvements could affect our findings. A uniformly better optimizer would likely preserve the observed scaling relationships while the actual optimal operating points might shift. In previous scaling law studies we do see the better optimizer can somehow smooth out the discontinuities in scaling behavior (Chen et al., 2023; Loshchilov & Hutter, 2019) or even enable new scaling regimes sometimes (e.g., LAMB (You et al., 2020) for large

---

**Algorithm 2** Generalized DP-SGD.

---

**Input:** Dataset $\mathcal{D}$, noise-batch ratio $\bar{\sigma}$, (expected) batch size $B$, iterations $T$
**Output:** Model parameters $\theta$.
Initialize model parameters $\theta_0 \in \mathbb{R}^M$
  **for** $t = 1$ **to** $T$ **do**
    Select a (possibly random) size $\approx B$ minibatch $\mathcal{B}_t \subset \mathcal{D}$
    $\bar{g} = \frac{1}{B} \sum_{\mathbf{x} \in \mathcal{B}_t} \text{clip}(\nabla \ell(\theta_{t-1}; \mathbf{x}))$
    $\tilde{g} = g + \bar{\sigma} \mathcal{N}(0, 1)^M$
    $\theta_t = \text{OptimizerUpdate}(\theta_{t-1}, \tilde{g})$
  **return** $\theta_T$

---

batch size pre-training shows a very different scaling behavior). The optimizers specifically designed for privacy-preserving training might recommend a new set of parameters to enable better absolute performance.

# B. Additional Details

## B.1. Notes on Generalized DP-SGD

**Minibatch Selection** We were vague in our description of the minibatch selection step. In most descriptions of DP-SGD, the minibatch is formed by Poisson subsampling with a fixed probability. Sampling with or without replacement, as well as deterministic batching are also possible (Balle et al., 2018). In our paper, we calibrated noise under both the Poisson sampling assumption and the deterministic batching strategy, picking the lower noise multiplier. When doing Poisson sampling, we use the sampling probability $B/N$ and noise multiplier $B \cdot \bar{\sigma}$.

**Known Quantities** If doing Poisson sampling, we typically are operating under the add/remove adjacency definition. Under this definition, $N$ is considered a sensitive quantity that we do not have access to directly, hence we cannot technically define the sampling probability as $B/N$ without violating DP. We also rely on $N$ later on, discussing its importance as it is interpreted as the data budget. If necessary, one can approximate $N$ quite accurately with DP since it is a simple count.

Alternatively, one can simply use the "zero-out" adjacency notion (Chua et al., 2024b), where $N$ is known but Poisson sampling still enjoys the same privacy analysis.

**Clipping Function** We omit a clipping norm parameter in the definition of "clip". This can be any function that maps an arbitrary real-valued vector to one with $\ell_2$-norm at most one. One standard choice is to clip the norm to $C$, and then divide by $C$ (De et al., 2022).

## B.2. Unit of Privacy and Multiple Participations

In traditional scaling laws work, it is common to assume access to an endless stream of data that does not require privacy protections. Therefore, every training example is only seen once, which simplifies the analysis of the scaling laws. In our case, we trained our models for $128K$ iterations with a physical batch size of $1024$, which is slightly less than a single pass over our entire dataset, satisfying the typical assumption. However, in our data analysis, we estimate what would happen with significantly larger batch sizes than we ran with, and in some cases this would involve multiple passes over the actual private dataset, something we did not account for directly in our analysis. Therefore, the actually setting that is best represented by our experimental methodology is not actually example-level DP, but rather user-level DP. There, we may assume that we have a finite number of users $N$ (which we should now interpret as the data budget), but we have an endless stream of data for each user. This circumvents the main concern, while allowing for users to participate multiple times during training which is typically very useful under DP. Alternatively, one can still consider the example-level DP setting, where each base example has multiple augmentations (e.g., rewritten text sequences that are semantically similar) that we can train on. All of our findings should hold, and be more reliable in this setting based on our methodology.

## B.3. FLOPs estimation under DP

As discussed, we approximate the compute cost $C$ as $6 \cdot M \cdot B \cdot S \cdot T$ based on the non-private scaling laws (Kaplan et al., 2020; Hoffmann et al., 2022) except that $B$ represents the number of examples (not tokens) in a batch, as this determines the privacy budget. This cost model is useful because we can directly compare to the non-private scaling laws. Further, this cost model is also accurate because the extra overhead of DP-SGD compared to Adam can be directly amortized: compiler-based

systems like GSPMD (Xu et al., 2021) and parallel machine learning libraries (Rush et al., 2024) let us parallelize the per-example gradient computations without a linear (in $B$) increase in memory usage. The total clipping costs are only a small linear cost (comprising of only element-wise operations and no matrix multiplications) in $M$, $T$, and $B$ (and are independent of sequence length $S$); the total noising costs are independent of $B$ and are linear in only $M$ and $T$. Thus, the overall compute in DP-SGD is dominated by the non-private approximation above.

## C. Additional Experiments

### C.1. Saturating Compute Budget

Building on our findings above, it is natural to ask where the saturation point occurs for different privacy budget and data budgets. This can be helpful to determine how much compute is needed to get the most utility under a fixed data and privacy budget, as well as how to spend that compute optimally. These results are shown in Table 2.

- With a higher data and privacy budget, we benefit substantially from larger compute budgets.
- With DP, the compute-optimal training configurations requires training significantly smaller models over significantly more tokens than without DP. For these training configurations, the ratio of training tokens to model parameters varies in different settings, but in all settings it is significantly larger than it would be without DP, where prior work found $20\times$ to be a good rule of thumb (Hoffmann et al., 2022).

*Table 2.* Saturating compute budgets, as well as optimal training configurations for those compute budgets across a representative set of data and privacy budgets.

| Data Budget | Privacy Budget | Compute Budget | Cross Entropy | Model Size | Iterations | Batch Size | Token / Model Ratio |
|---|---|---|---|---|---|---|---|
| $1.0 \times 10^5$ | 1 | $1.3 \times 10^{16}$ | 7.28 | $4.6 \times 10^6$ | $1.8 \times 10^3$ | $5.1 \times 10^2$ | $1.0 \times 10^2$ |
| | 4 | $1.1 \times 10^{17}$ | 6.65 | $4.6 \times 10^6$ | $1.8 \times 10^3$ | $4.1 \times 10^3$ | $8.5 \times 10^2$ |
| | 16 | $2.0 \times 10^{18}$ | 5.60 | $1.7 \times 10^7$ | $2.7 \times 10^3$ | $1.4 \times 10^4$ | $1.1 \times 10^3$ |
| | 64 | $7.5 \times 10^{18}$ | 4.63 | $2.0 \times 10^7$ | $6.3 \times 10^3$ | $1.9 \times 10^4$ | $3.2 \times 10^3$ |
| $1.0 \times 10^6$ | 1 | $2.8 \times 10^{17}$ | 5.89 | $4.6 \times 10^6$ | $2.5 \times 10^3$ | $8.2 \times 10^3$ | $2.3 \times 10^3$ |
| | 4 | $8.8 \times 10^{18}$ | 4.62 | $1.9 \times 10^7$ | $6.5 \times 10^3$ | $2.3 \times 10^4$ | $4.1 \times 10^3$ |
| | 16 | $3.3 \times 10^{19}$ | 3.61 | $1.7 \times 10^7$ | $1.4 \times 10^4$ | $4.6 \times 10^4$ | $1.9 \times 10^4$ |
| | 64 | $3.2 \times 10^{20}$ | 2.82 | $4.9 \times 10^7$ | $1.2 \times 10^4$ | $1.9 \times 10^5$ | $2.2 \times 10^4$ |
| $1.0 \times 10^7$ | 1 | $3.8 \times 10^{19}$ | 3.73 | $1.7 \times 10^7$ | $9.6 \times 10^3$ | $7.8 \times 10^4$ | $2.3 \times 10^4$ |
| | 4 | $3.8 \times 10^{20}$ | 2.81 | $4.9 \times 10^7$ | $1.1 \times 10^4$ | $2.2 \times 10^5$ | $2.6 \times 10^4$ |
| | 16 | $2.0 \times 10^{21}$ | 2.15 | $7.0 \times 10^7$ | $1.2 \times 10^4$ | $7.4 \times 10^5$ | $6.7 \times 10^4$ |
| | 64 | $4.4 \times 10^{22}$ | 1.66 | $3.3 \times 10^8$ | $4.9 \times 10^4$ | $8.8 \times 10^5$ | $6.7 \times 10^4$ |
| $1.0 \times 10^8$ | 1 | $5.2 \times 10^{21}$ | 2.26 | $1.3 \times 10^8$ | $5.8 \times 10^4$ | $2.2 \times 10^5$ | $4.9 \times 10^4$ |
| | 4 | $4.4 \times 10^{22}$ | 1.66 | $3.3 \times 10^8$ | $4.9 \times 10^4$ | $8.8 \times 10^5$ | $6.7 \times 10^4$ |
| | 16 | $1.0 \times 10^{23}$ | 1.32 | $3.3 \times 10^8$ | $9.3 \times 10^4$ | $1.0 \times 10^6$ | $1.5 \times 10^5$ |
| | 64 | $1.0 \times 10^{23}$ | 1.23 | $3.3 \times 10^8$ | $1.1 \times 10^5$ | $8.8 \times 10^5$ | $1.5 \times 10^5$ |
| $1.0 \times 10^9$ | 1 | $8.5 \times 10^{22}$ | 1.36 | $3.3 \times 10^8$ | $9.4 \times 10^4$ | $8.8 \times 10^5$ | $1.3 \times 10^5$ |
| | 4 | $1.0 \times 10^{23}$ | 1.23 | $3.3 \times 10^8$ | $1.1 \times 10^5$ | $8.8 \times 10^5$ | $1.5 \times 10^5$ |
| | 16 | $1.0 \times 10^{23}$ | 1.22 | $3.3 \times 10^8$ | $1.1 \times 10^5$ | $8.8 \times 10^5$ | $1.5 \times 10^5$ |
| | 64 | $1.2 \times 10^{23}$ | 1.20 | $3.3 \times 10^8$ | $1.1 \times 10^5$ | $1.1 \times 10^6$ | $1.8 \times 10^5$ |

### C.2. Full Experiment Grid

In Figure 6, we plot the cross-entropy loss for different privacy budgets, data budgets, and compute budgets under varying numbers of iterations, model sizes, and batch sizes. Much can be learned from these plots, including:

- The optimal number of iterations typically falls around $T \approx 10K$, and the optimal batch size often falls in the range $B \approx 10 - 100K$, although neither of these is universally true and as expected it depends on the values of the privacy, data, and compute budgets. Batch size seems to be the most important parameter, as indicated by the steep slope of those lines.

### C.3. Physical Batch Size Ablation

Central to our methodology is an assumption that for a fixed noise-batch ratio, the training curves should be similar for different physical batch sizes. In this section, we conduct ablations to test this hypothesis, and quantify the impact of varying physical batch size under a fixed noise-batch ratio. We consider 3 values for noise-batch ratio: $0.5^{20}$, $0.5^{15}$, and $0.5^{10}$, and

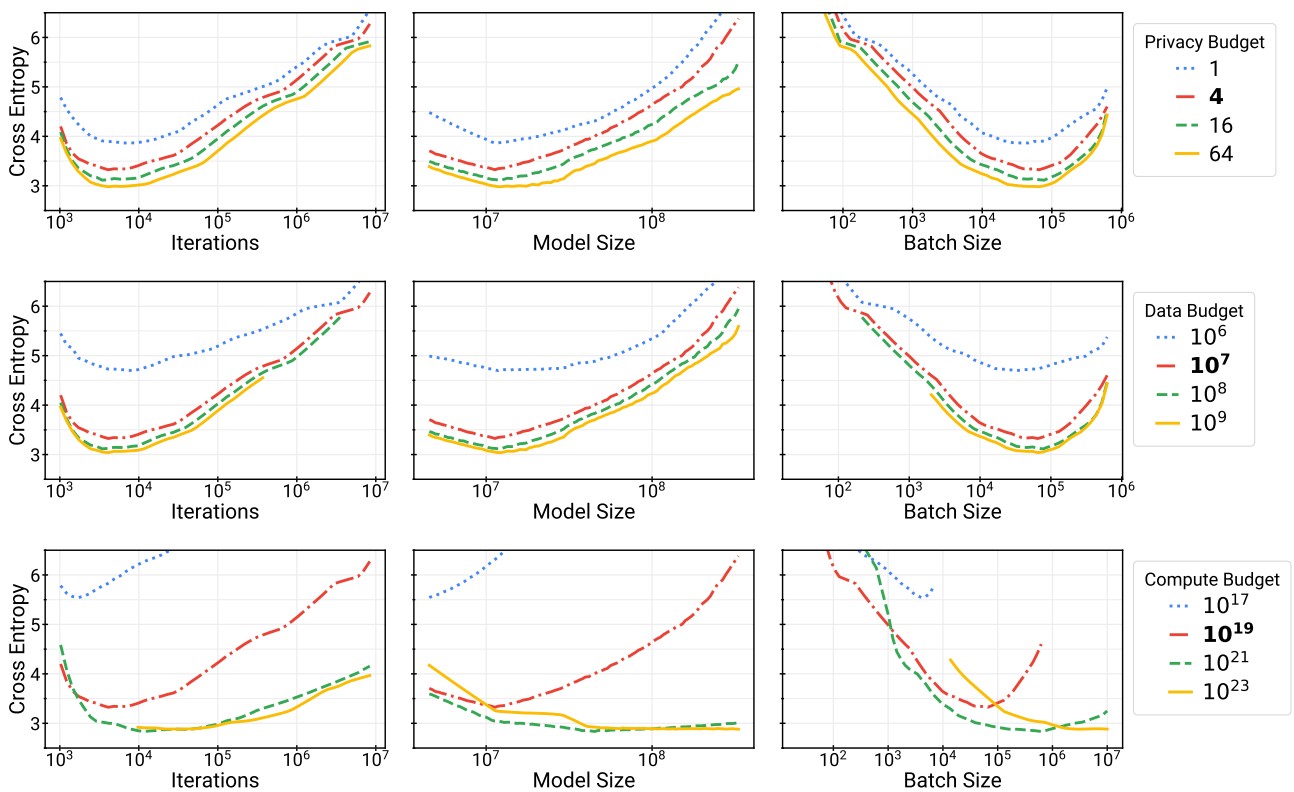

*Figure 6.* Cross-entropy of best models trained in each setting. From top to bottom , we vary the Privacy Budget, Data Budget, and Compute Budget, keeping the other two budgets fixed to default values (bolded). From left to right, we vary the number of Iterations, the Model Size, and the Batch Size, and treat the other two as nuisance parameters which we minimize over.

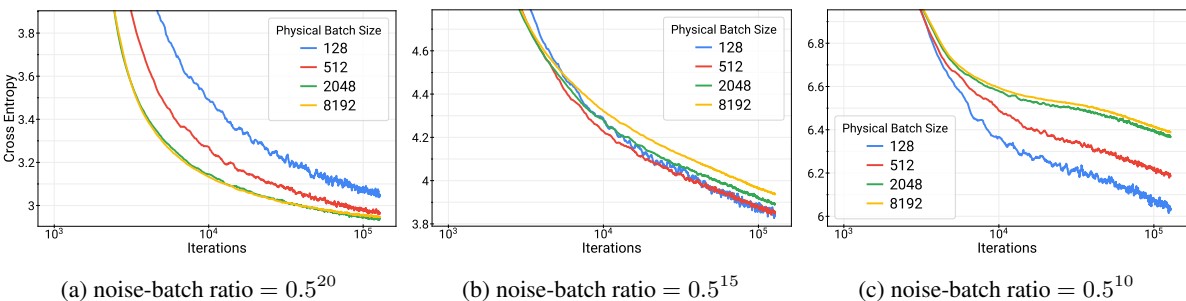

*Figure 7.* Cross-entropy loss of BertTiny averaged over 3 trials for different physical batch sizes and noise-batch ratio values.

physical batch sizes of 128, 512, 2048, and 8192. For this ablation we focus on the BertTiny model, which we train for 128K iterations. We average the losses across three random trials.

The results of this experiment are shown in Figure 7. Our primary findings are:

- At the smallest noise-batch ratio in Figure 7a, results are as expected. Specifically, larger batch sizes lead to better model performance, but there are diminishing returns. Physical Batch Sizes of 2048 and 8192 have nearly identical training curves.
- At the medium and larger noise-batch ratio values shown in Figures 7b and 7c, we observe a surprising phenomenon: smaller physical batch sizes lead to models with lower loss. The effect is most prominent in Figure 7c. We do not have a good explanation for this behavior, but we did additional experiments to rule out some plausible explanations in Appendix C.4. Large physical batch sizes ($B = 2048$ and $B = 8192$) still have very similar learning curves.

While the results of this experiment did not fully match expectations, a similar behavior was observed in prior work (Sander

et al., 2023) (Figure 4b). Moreover, for sufficiently large batch sizes the training curves are very similar across all noise-batch ratio values tested. Thus, we believe that the physical batch size of $1024$ that we use in our main experiments is a reasonable (although not perfect) indicator of what would happen with much larger batch sizes that would be needed to get favorable privacy/utility trade-offs in real-world settings. Understanding when and why thsi behavior manifests is a very interesting direction for future work.

### C.4. Physical Batch Size Ablation - Extended

In Appendix C.3 we observed a surprising phenomenon where for some values of noise-batch ratio, smaller physical batch sizes perform better than larger physical batch sizes. This is in contrast to our initial hypothesis, and our experimental results for very small values of noise-batch ratio that larger physical batch sizes should be on par with or better than smaller physical batch sizes for the same noise-batch ratio.

While we do not have a great explanation for the observed phenomenon, we have ruled out several possible explanations, which we discuss below:

1. **Learning Rate Tuning**. While our main experiment used a fixed learning rate of $0.5^8$ across all values of noise-batch ratio, we ran further experiments for a noise-batch ratio of $0.5^{15}$ with four different learning rates ($0.5^6, 0.5^7, 0.5^8, 0.5^9$), and report the best cross-entropy across all learning rates on a per-iteration basis. Even with learning rate tuning, the conclusion is the same: smaller physical batch sizes achieve lower loss than larger ones (see Figure 8).

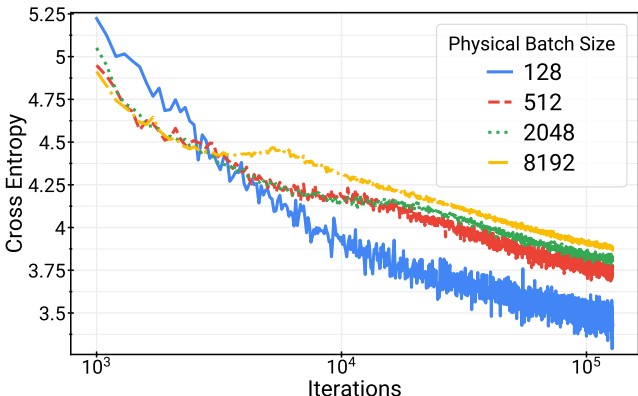

*Figure 8.* Smaller physical batch sizes achieve lower loss than larger ones.

2. **Differences in Train / Eval Loss**. Our main experiment measures the training loss, but since the loss is computed before incorporating the gradient into the model, and because we train for less than one pass over the entire dataset, this is an unbiased estimate of the evaluation loss. It is natural to ask which models have lower final loss on the training set (after incorporating those examples into the model). To test whether lower physical batch sizes somehow generalize better, or whether they also do better on the training loss, we measured the loss of the final trained model on 1M examples from the training set. We focus on the noise-batch ratio of $0.5^{15}$ in this test. The table below shows that smaller physical batch sizes also have better performance on the already-seen training examples, ruling out this explanation (see Table 3).

| Batch Size | Training Set | |
| --- | --- | --- |
| | Cross-Entropy | Accuracy |
| 128 | 3.586 | 43.59% |
| 512 | 3.971 | 37.27% |
| 2048 | 4.01 | 37.55% |
| 8192 | 4.057 | 36.73% |

*Table 3.* Loss over the entire training set is also better for lower physical batch sizes.

3. **Model Size**. The main experiment uses BertTiny, which is a relative small model. It is natural to ask whether the same behavior would be observed for a larger model like BertBase. The figure below shows that the same phenomenon happens for BertLarge, but only for the largest noise-batch ratio. The other two values of noise-batch ratio do not

exhibit this behavior, although at the middle noise-batch ratio, the trend line suggests there may be a crossover point beyond the limits of the x axis. Thus, increasing model size seem to influence and mitigate this behavior, but not eliminate it completely. See Figure 9.

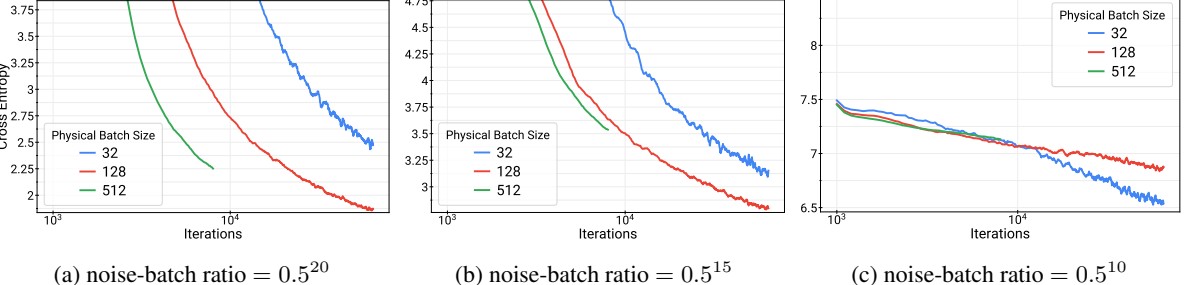

(a) noise-batch ratio $= 0.5^{20}$      (b) noise-batch ratio $= 0.5^{15}$     (c) noise-batch ratio $= 0.5^{10}$

*Figure 9.* Cross-entropy loss of `BertLarge` averaged over 3 trials for different physical batch sizes and noise-batch ratio values.

4. **Training Pipelines**. It is natural to question whether this behavior is explained by some bug in the training pipeline. We carefully reviewed the implementation and did not find any bugs that could explain this behavior, and also did additional experiments on a totally separate training pipeline based on `NanoDO` (Liu et al., 2024), where we observed the same qualitative behavior when training a 30M parameter decoder-only transformer model with `DP-Adam` for 32K iterations. The figures below show the smoothed cross-entropy averaged over 3 random trials.

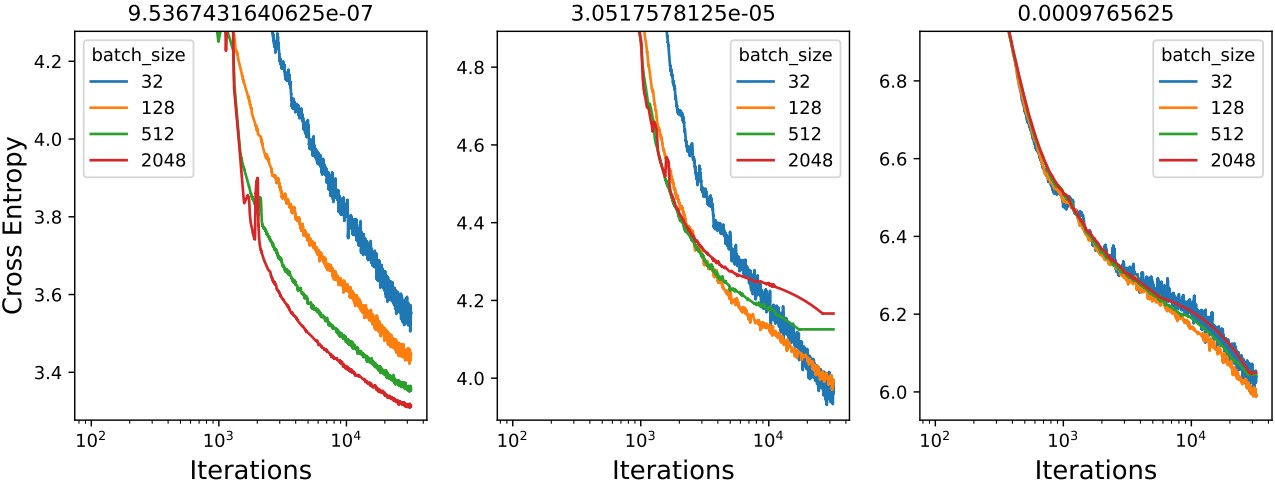

*Figure 10.* Loss on NanoDO (Liu et al., 2024).

## C.5. Training Throughput

By looking at intermediates, using a single physical batch size, and separating the accounting from the experimentation we greatly reduce the number of experiments to run. However, the set of experiments we outline above is still very compute-intensive. We utilize TPUv3 pods to run all experiments, and configured the models to use pure data parallelism, using more cores for larger models so that each experiment finishes within four to ten hours. `BertTiny` was trained on 16 TPUv3 cores, while `BertLarge` was trained on 128. Table 4 provides the training throughputs for all models in our experiments.

| Model | Params | Steps/sec | Per Core Batch Size | Records / Sec |
|---|---|---|---|---|
| BertTiny | 4.52M | 8.959 | 64 | 573 |
| BertMini | 11.4M | 5.494 | 64 | 352 |
| BertSmall | 29.0M | 6.602 | 32 | 211 |
| BertMedium | 41.6M | 4.196 | 32 | 134 |
| BertBase | 110M | 3.621 | 16 | 54 |
| BertLarge | 335M | 2.225 | 8 | 17.8 |
| BertMega | 729M | 1.536 | 4 | 6.1 |

*Table 4.* Training throughput for various BERT models

## C.6. Reproducing non-private scaling laws results

We now confirm that the experimental data we collected matches the expected behavior of Hoffmann et al. (2022), specifically that in the absence of noise, the optimal model size and tokens should grow in roughly equal proportion with increasing compute budget. This is true despite our several methodological differences, including: (1) doing per-example gradient clipping, (2) using a different optimizer and not retraining for each number of iterations, (3) using a large physical batch size, etc. The exact Token / Model ratio predicted here is larger than prior work, but that is well explained by the fact that a batch size of 1024 examples is well beyond the critical batch size of compute-efficient training (McCandlish et al., 2018).

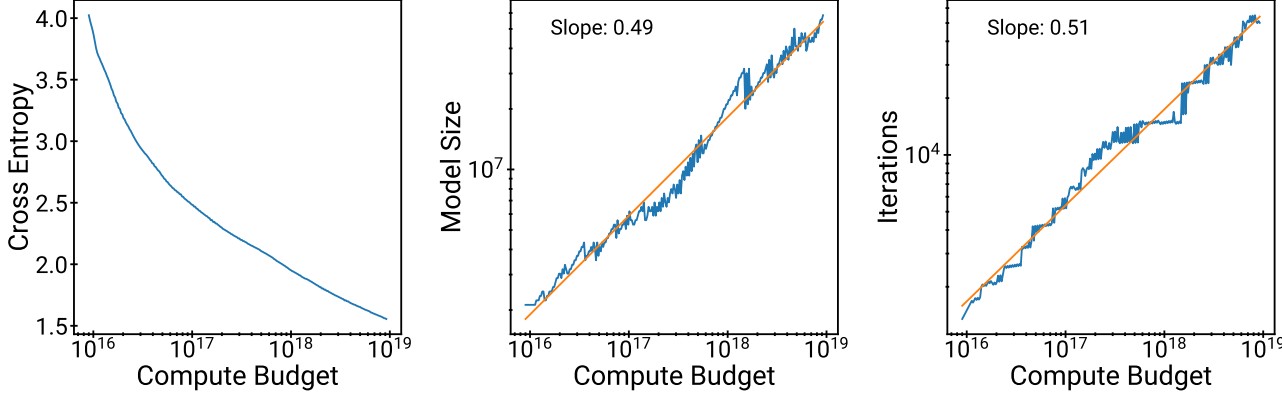

*Figure 11.* Compute-optimal cross-entropy, model size, and number of iterations when running DP-Adam with $\sigma = 0$.

## C.7. Optimal Learning Rates

We now look at the training curves for different learning rates and different noise-batch ratio values. These results generally match expectations and demonstrate that the learning rates we chose were selected from the correct regime.

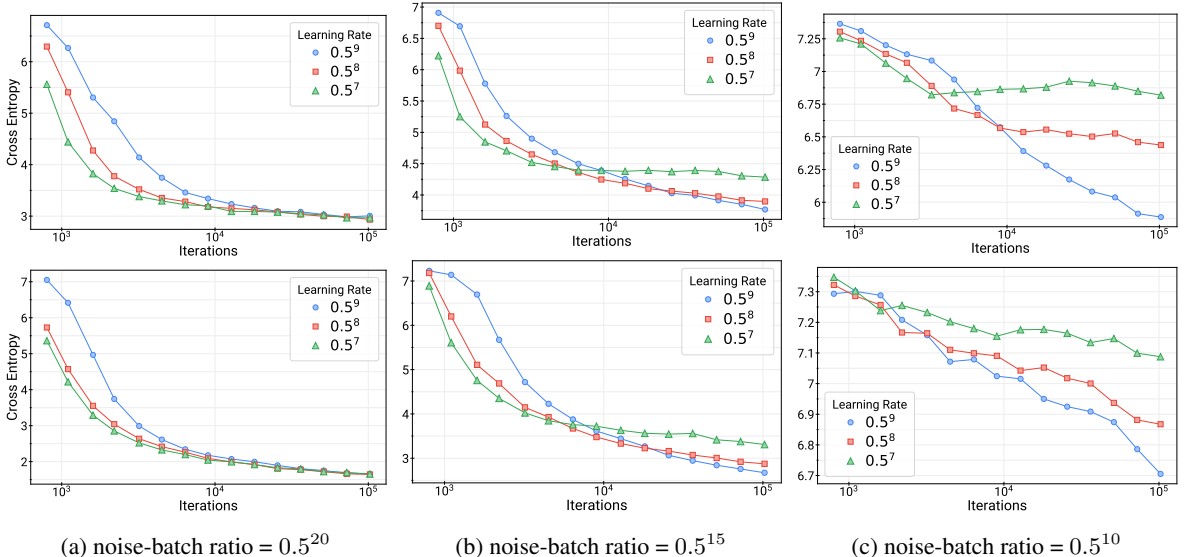

(a) noise-batch ratio = $0.5^{20}$      (b) noise-batch ratio = $0.5^{15}$      (c) noise-batch ratio = $0.5^{10}$

*Figure 12.* Training curves for BertTiny (top) and BertMedium (bottom) with varying learning rates at different noise-batch ratio values.

### C.8. Optimal Compute Budget Allocation

In this section, we extend the results from Section 4.1, including results for more settings of the data budget, ranging from $N = 10^6$ to $N = 10^9$. The full results are shown in Figure 13. Our findings are qualitatively similar to the ones we identified in the main text across different data budgets, but the precise constants may differ.

### C.9. Smoothing and Extrapolation

In Figure 14 we visualize how our semi-parametric smoothing approach works. Since each raw measurement is an average cross-entropy over $1024 \cdot 100$ examples, it is naturally a noisy estimate of the "true" cross-entropy. Our smoothing strategy ensures the appropriate monotonicity properties are enforced, while matching the overall trend as closely as possible.

## D. Caveats on Privacy Calibration

Throughout the work, we have assumed that hyperparameter choices for model training are made against a *fixed* privacy budget. In particular, we assume the common scenario in which the model trainer fixes an $(\epsilon, \delta)$-budget and then utilizes a *privacy calibration* algorithm to choose DP-SGD hyperparameter combinations (sampling probability, training iterations and noise scale) which satisfy this privacy budget. Note that in the main manuscript, we express this choice in terms of the noise-batch ratio $\sigma$ and the number of iterations $T$, but this is merely a matter of notation. As also noted in the preceding subsection, the choice of sampling probability (and thus the resulting batch size) play an important role in determining the final model's cross-entropy. As described in the recent work of Kaissis et al. (2024), calibrating against a fixed $(\epsilon, \delta)$-budget while varying DP-SGD hyperparameters must be done with care: In brief, one cannot assume that DP-SGD with different hyperparameters offers the same privacy guarantees *despite* having the same nominal $(\epsilon, \delta)$-budget. This is due to the fact that the privacy guarantees of DP-SGD can only be adequately expressed through a *privacy profile*, that is, a collection of $(\epsilon, \delta(\epsilon))$ tuples. In simple terms, two DP-SGD algorithms can share an $(\epsilon, \delta)$-budget, that is, offer the same privacy guarantees for a specific $\delta$ while offering (sometimes drastically) different privacy guarantees at a different value of $\delta$. As also described in the aforementioned work, varying the sampling rate (and thus batch size) has a drastic impact on this difference in privacy guarantees. The authors of the aforementioned work thus recommend reporting the *excess vulnerability* that DP-SGD algorithms incur with respect to each other when they replace one another in a workflow. We refer to the aforementioned work for technical details. Here, we demonstrate that meaningful differences can indeed arise between models calibrated to satisfy the same $(\epsilon, \delta)$-budget.

Exemplarily, we fixed a privacy budget of $(\epsilon, \delta) = (8, 10^{-8})$ for specific fixed compute budgets and model sizes while varying the batch size (and adjusting the noise to maintain the privacy budget). We then computed the scaling-law predicted cross-entropy and the vulnerability of the models against membership inference attack (MIA) adversaries measured in

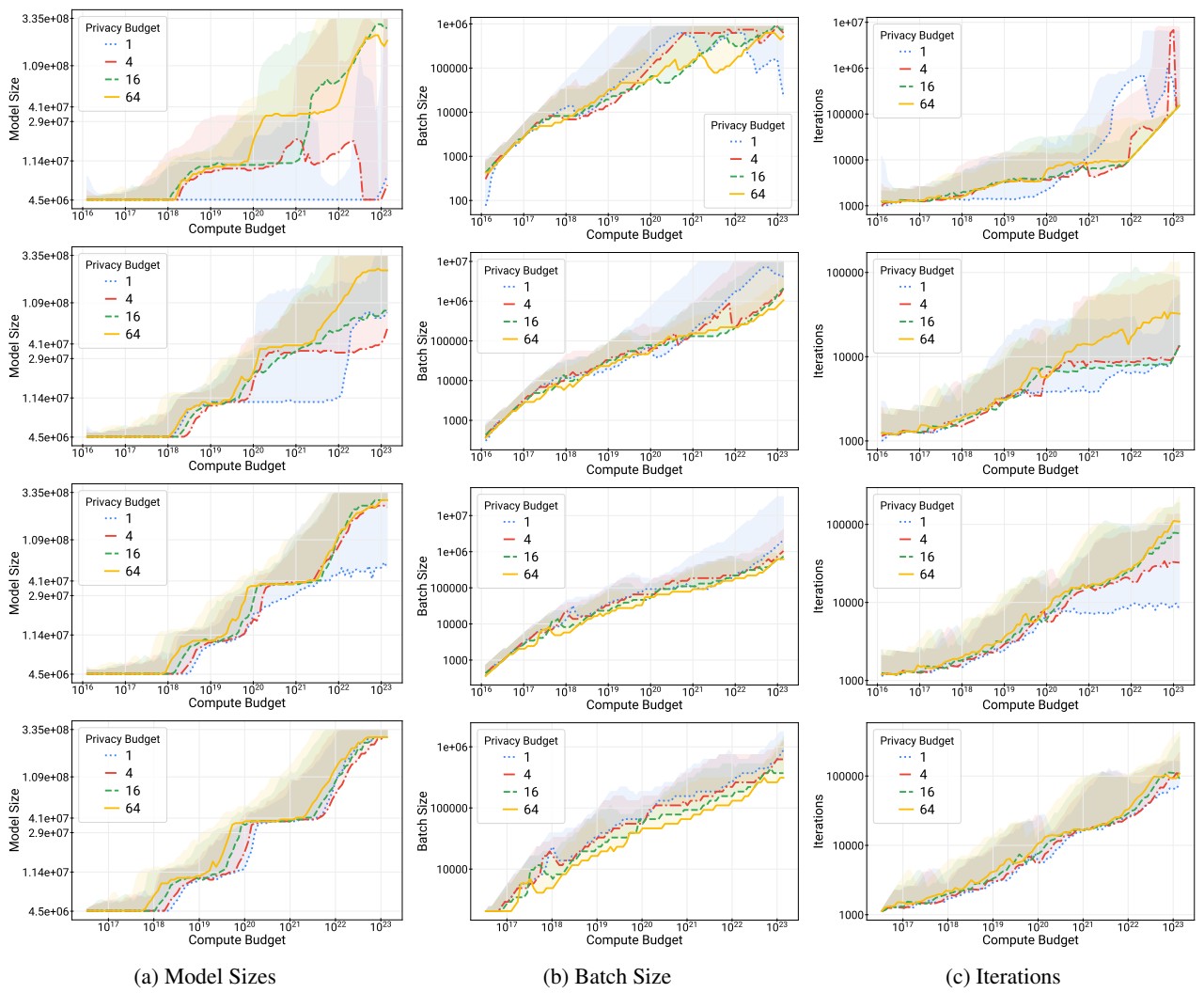

(a) Model Sizes  (b) Batch Size  (c) Iterations

*Figure 13.* Compute optimal model-sizes, batch sizes, and iterations for varying privacy budgets and compute budgets, and data budgets. Each row of plots corresponds to a different data budget of $N = 10^6, 10^7, 10^8$, and $10^9$ respectively. Each line corresponds to the minimum value of that hyper-parameter that achieves within 1% of the optimal cross-entropy across all constant-compute training configurations. The shaded region corresponds to the full range of possible values for that hyper-parameter that are optimal to within 1%.

terms of MIA *advantage* (Yeom et al., 2018). We note that MIA advantage is a proxy metric for other attacks such as reconstruction attacks and is related to the $\Delta$-divergence which quantifies vulnerability as described in Kaissis et al. (2024). Figure 15 demonstrates the phenomenon.

Note that in all three cases, it is possible to achieve virtually the same cross-entropy (blue, left vertical axis) while controlling the MIA advantage by judiciously choosing the batch size. Conversely, it is also possible to incur an unduly high vulnerability without a substantial decrease (or sometimes even an increase) in cross-entropy through a poor choice of batch size. As an auxiliary finding, we note that the relationship between cross-entropy and batch size follows the trend observed in De et al. (2022). In brief, there is a *Pareto optimal* batch size beyond which both the cross-entropy *and* the excess vulnerability can only become worse (larger). We stress that the models shown here all satisfy the *same nominal* $(\epsilon, \delta)$-*budget* but exhibit (substantial) differences in vulnerability against at least a subset of adversaries which may pass unnoticed if only reporting a single $(\epsilon, \delta)$-DP guarantee. We thus recommend practitioners to monitor changes in excess vulnerability that may arise due to hyperparameter tuning and report them alongside the $(\epsilon, \delta)$-budget to which DP-SGD has been calibrated.

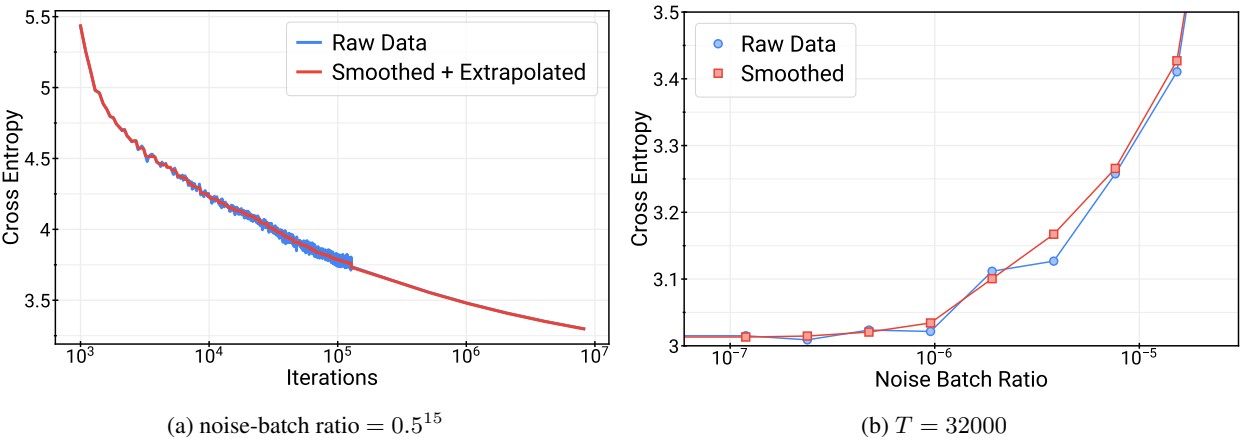

(a) noise-batch ratio $= 0.5^{15}$

(b) $T = 32000$

*Figure 14.* Demonstration of our semi-parametric smoothing on BertTiny.

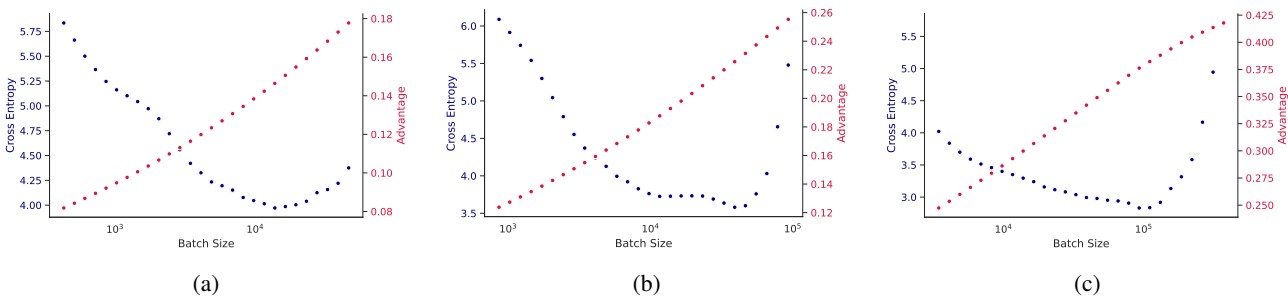

(a)

(b)

(c)

*Figure 15.* Varying the batch size (horizontal axis, log-scale) has a drastic effect on excess vulnerability (measured as MIA advantage, red, right vertical axis) for models with a fixed compute budget and size and a fixed privacy budget of $(\epsilon, \delta) = (8, 10^{-8})$. (a): Compute budget: $6 \cdot 10^{17}$, model size: $4\,000\,000$. (b) Compute budget: $6.3 \cdot 10^{19}$, model size: $200\,000\,000$. (c) Compute budget: $2.5 \cdot 10^{20}$, model size: $200\,000\,000$. The scaling-law-predicted cross-entropy is plotted on the left vertical axis in blue.

## E. Parametric Scaling Laws

Previous work on (non-private) LLM scaling laws use a fully parametric form to predict the cross-entropy loss based on several key factors. For example, the "Chinchilla" scaling law (Hoffmann et al., 2022) can be parameterized as follows:

$$\hat{L}(n_{\text{params}}, n_{\text{tokens}}) \triangleq E + \frac{A}{n_{\text{params}}^\alpha} + \frac{B}{n_{\text{tokens}}^\beta}.$$

In this section, we explore a similar methodology to fit a fully parametric form of scaling law in the setting of private training. Following the notation of this paper, we define a parametric form based on the following key factors: the model size $M$, the number of examples $N$ and the noise-batch ratio $\bar{\sigma}$. Note our notations are slightly different from Hoffmann et al. (2022), and we use number of *examples* instead of number of *tokens* as it is a more relevant quantity in private training.

We consider several variations of parametric forms. The first one is a naive extension of the Chinchilla scaling law, by adding an additional term involving the noise-batch ratio:

$$\hat{L}_{\mathbf{1}}(M, N, \bar{\sigma}) \triangleq E + \frac{A}{M^\alpha} + \frac{B}{N^\beta} + C\bar{\sigma}^\gamma. \tag{1}$$

We did not put $\bar{\sigma}^\gamma$ in the denominator because the loss increases with the noise-batch ratio. Following Hoffmann et al. (2022), we estimate the coefficients $(E, A, B, C, \alpha, \beta, \gamma)$ by minimizing the Huber loss (Huber, 1992) between the predicted and the observed loss using the L-BFGS algorithm (Nocedal, 1980), and we try multiple different initializations and choose the best fit. We restrict the curve fitting data to only the subsets of data points with more than $100,000$ training iterations, noise-batch ratio larger than $5 \times 10^{-7}$, and ignore points with very high cross-entropy loss ($> 8$).

Figure 16 shows the optimal fit. We observe that the prediction is generally accurate for low loss value ranges. However, the prediction starts to diverge at high loss value ranges, corresponding to runs with high noise-batch ratio. This is partly due to

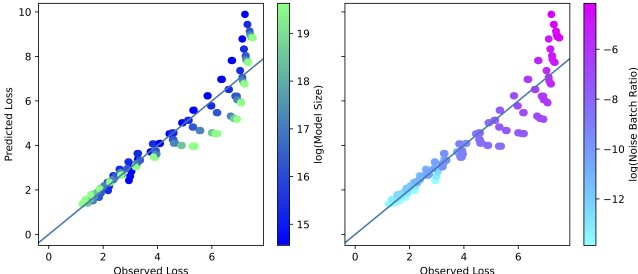

*Figure 16.* Parametric private scaling law of $\hat{L}_1$ from Equation (1). Optimal fit with $\alpha = 0.71$, $\beta = 12.87$, $\gamma = 0.19$. The two pannels show the same plot of observed cross-entropy loss against the predicted loss from the scaling law, except the data points are colored differerently, according to the model size and noise-batch ratio, respectively.

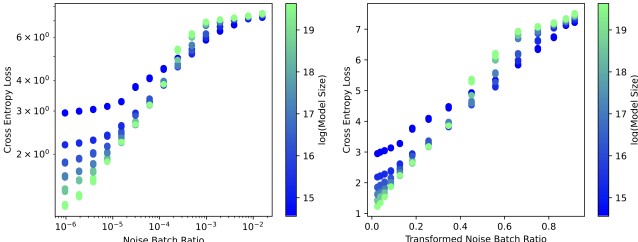

*Figure 17.* Relation between the noise-batch ratio and the cross-entropy loss. **(left)** The data plotted in log-log scale. **(right)** The data plotted in linear scale, where the noise-batch ratio $\bar{\sigma}$ is transformed according to a simple rule in Equation (2).

the fact that the noise-batch ratio does not impact the loss in a log-linear fashion, as shown on the left panel of Figure 17. Therefore, the parametric form of Equation (1) cannot capture the relation accurately. Instead, we observe S-shaped curves in the log-log plot. To account for this, we apply a simple transform to the noise-batch ratio $\bar{\sigma}$:

$$\bar{\sigma}_{\curvearrowright} \triangleq \text{sigmoid}\left(\frac{\log(\bar{\sigma}) + 8}{1.6}\right). \tag{2}$$

The right panel of Figure 17 shows an approximately linear relation after this transformation. Furthermore, we observe that the relation between the noise-batch ratio and the loss changes with the model sizes.

After incorporating those observations, we consider an alternative variant of private scaling law parameterization:

$$\hat{L}_2(M, N, \bar{\sigma}) \triangleq E + \frac{A}{M^\alpha} + \frac{B}{N^\beta} + \frac{C\bar{\sigma}_{\curvearrowright}^\gamma}{M^{\alpha_2}}. \tag{3}$$

The optimal fit according to this parameterization is shown in Figure 18. We observe that the predicted loss matches with the observed loss better than the previous parameterization in Figure 16.

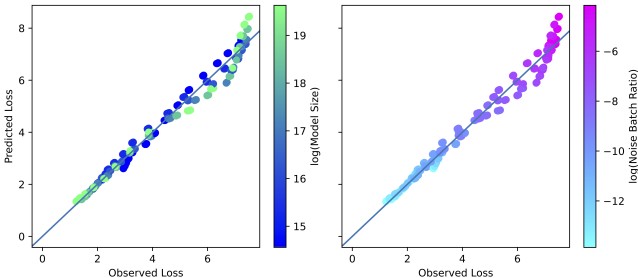

*Figure 18.* Parametric private scaling law of $\hat{L}_2$ from Equation (3). Optimal fit with $\alpha = 0.47$, $\beta = 0.12$, $\gamma = 0.95$, $\alpha_2 = -0.07$. The two pannels show the same plot of observed cross-entropy loss against the predicted loss from the scaling law, except the data points are colored differerently, according to the model size and noise-batch ratio, respectively.

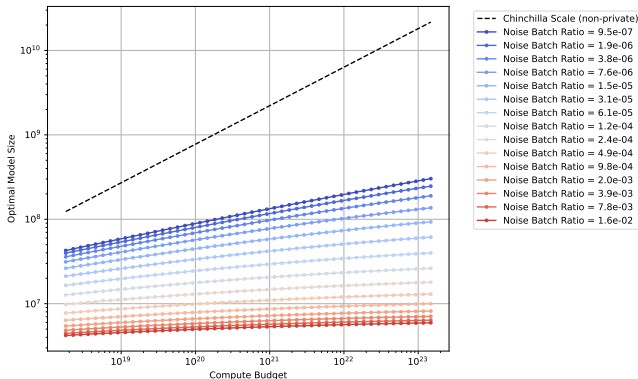

*Figure 19.* Optimal model sizes under according to the parametric private scaling law in Equation (3).

In the Chinchilla parameterization of scaling law for non-private LLMs, the optimal model size under a certain compute budget (approximately represented by $6n_{\text{params}}n_{\text{tokens}}$) can be directly solved and takes a power-law form (Hoffmann et al., 2022, Equation (4)). In our case, the parameterization is more complicated, for a given compute budget and noise-batch ratio, we use `scipy.optimize.minimize_scalar` to find the optimal model size that minimizes $\hat{L}_2$. The results are plotted in Figure 19. We observe that the slope is lower for curves with larger noise-batch ratio, indicating the challenges to scale model sizes under heavy DP noises. As the noise decreases, the curves shift up and the slopes increase, approaching towards the non-private Chinchilla scaling law shown in dashed line.

While a fully parametric scaling law can be easier to interpret and understand, as noted above, there is not a simple log-linear relation between the loss and the noise-batch ratio. Our sigmoid based transformation (and the coupling with the model size) improved the tightness of the fitting. But the transformation is not designed in a very principled way. As a result, we opt to use the semi-parametric fitting in Section 3 in the main analysis of our results. We also leave the exploration of other alternative parametric fitting such as fitting a $\bar{\sigma}$-depending delta term on top of a non-private scaling law for future work.

