# OpenReview forum: "Scaling Laws for Differentially Private Language Models"
_ICML.cc/2025/Conference — ICML 2025 poster_

### Official Review · Reviewer_U13Y · 2025-02-15

**Overall Recommendation:** 1

**Summary:**

This paper formulates scaling laws that accurately reflect the complexities of training Differentially Private (DP) Large Language Models (LLMs).

**Claims And Evidence:**

Yes, the claims made in the submission are supported by clear and convincing evidence.

**Essential References Not Discussed:**

No.

**Experimental Designs Or Analyses:**

Yes, I checked the soundness/validity of any experimental designs or analyses.

**Methods And Evaluation Criteria:**

It is not clear whether the proposed methods make sense for the problem or application at hand.

**Other Comments Or Suggestions:**

1. As for Figure 1, could you polish the y-axis? For example, Figure 1(b) sometimes uses scientific notation, and sometimes does not use it.
2. As for Figure 1(b), I guess that you try to vary the batch size, but the legend shows that you try to vary the privacy budget.
3. As for Figure 3(b), it is not clear to me that the compute budget increases when you fix the data budget and model size. My understanding is that FLOPs = 6ND, where N is the model size and D is the number of tokens.
4. As for Table 2, we usually use 35M to represent the model size instead of $3.5 \times 10^{7}$. We also use 512 for batch size and 1800 for iterations.

**Other Strengths And Weaknesses:**

Strengths:
1. The writing of this paper is excellent, and it is easy for me to follow this paper.
2. The authors conducted thorough experiments to gain insights into DP LLM training.

Weaknesses:
1. The motivation of this paper is not clear. I believe that OpenAI, Anthropic, XAI, Google, and DeepSeek are unlikely to use differential privacy for pre-training, as it may negatively impact model performance. Therefore, I am not sure whether we need such a scaling law in reality.
2. More and more people use decoder-only models such as Qwen or Llama. I am unclear why the authors chose to use Masked Language Modeling (BERT) in this paper.
3. Could you show me other papers that use BertMega model? From Huggingface, I did not find the config of the BertMega model.
4. Scaling laws are used to predict the behavior of large models. Therefore, I think the authors need to train a 1B-2B to evaluate the correctness of their scaling laws.
5. Furthermore, could you show me the results of your pre-trained models over downstream tasks?

**Questions For Authors:**

See strengths and weaknesses.

**Relation To Broader Scientific Literature:**

I believe the key contribution of the paper lies in conducting numerous experiments.

**Theoretical Claims:**

This paper does not include any proof.

---

> ### Author Rebuttal · Authors · 2025-04-01
>
> We appreciate the feedback, and thank you for sharing ideas for specific ways to improve the paper. We are glad that you found the paper well-written and thorough, but we respectfully disagree with some of your comments, and respond to the main critiques below.
>
> **[Motivation]** There are several complementary reasons why we believe DP scaling laws for pretraining are well-motivated and of interest to the privacy community.
>
> * *Importance of pretraining privacy*: For current frontier models, and pretraining specifically, there is increasing need for data privacy, e.g., due to risks of memorization in large models. This makes DP pretraining research relevant as pretraining data is often scraped from the web (and can contain sensitive personal information [1]). The need for privacy will further increase if companies start training on sensitive user data as they run out of public datasets to train on.
>
> * *Addressing utility challenges*: Utility degradation due to DP actually serves to motivate our work. With enough data and compute, and well-tuned mechanisms, the performance degradation due to DP can be mitigated to a large degree [2]. We believe scaling laws will help in this mitigation.
>
> * *Broader Context*: Privacy-preserving machine learning falls squarely within the [ICML call for papers](https://icml.cc/Conferences/2025/CallForPapers) under the Trustworthy Machine Learning category, and the best paper from ICML 2024 highlighted DP pre-training as an important future direction [1]. Several large organizations have invested in this space already [2,3,4,5,6].
>
> [1] Tramer et al., Position: Considerations for Differentially Private Learning with Large-Scale Public Pretraining
>
> [2] De et al., Unlocking High-Accuracy Differentially Private Image Classification through Scale
>
> [3] Xu et al., Federated Learning of Gboard Language Models with Differential Privacy
>
> [4] Anil et al., Large-Scale Differentially Private BERT
>
> [5] Pelikan et al., Federated Learning With Differential Privacy for End-to-End Speech Recognition
>
> [6] https://www.microsoft.com/en-us/research/group/privacy-preserving-machine-learning-innovation/
>
> **[Encoder-decoder Models vs. Decoder-only models]**
> Our choice to study BERT models in this research was motivated by previous research on pre-training with differential privacy [3]. We agree that our focus on BERT models raises questions about generality to other model architectures. We acknowledged this limitation in Appendix A. We further provide evidence in Section 3.7 that despite the differences in training setups with prior (non-private) scaling laws work, we are able to reproduce the main finding of Hoffman et al., i.e., that data and model size should be increased in equal proportion with increasing compute. This provides some evidence that our results should translate to other settings. This observation is also consistent with [7] who found qualitatively similar scaling between encoder-decoder and decoder-only architectures. See also our response to Reviewer qUEy.
>
> [7] Yao et al., Towards Neural Scaling Laws for Time Series Foundation Models
>
> **[BertMega]** BertMega is the only non-standard model config we used, to also explore a larger BERT model than the ready-made configs listed at https://github.com/google-research/bert. We will note this explicitly in the revision.
>
> **[1B-2B Models]** This is a good idea, and something we discuss in the limitations section (Appendix A). Scaling to larger models introduces significant engineering challenges, most notably handling model parallelism. For such models, weights—and thus their corresponding gradients—are typically sharded across devices. The gradient clipping step in DP-SGD requires storing and synchronizing per-example gradients across layers and devices for clipping, which necessitates substantial changes to existing implementations. While some work has explored this in the context of LLM fine-tuning [8], these challenges remain for training LLMs from scratch. We will be sure to highlight this as an additional challenge to scale DP training to the billion-parameter scale (in addition to DP scaling laws) in the revision.
>
> [8]: He et al., Exploring the Limits of Differentially Private Deep Learning with Group-wise Clipping
>
> **[Evals on downstream tasks]** Thank you for raising this point. We note that performance on downstream tasks are generally too noisy for scaling law studies themselves, as a result, we follow the previous work on LLM scaling laws and focus on the crossentropy loss. We agree that beyond the scaling law studies, it would be valuable to show some downstream evaluations on large (maybe 1B-2B models) DP pre-trained models. We will highlight this as an important direction for future work in revision.
>
> **[Other suggestions]** We will incorporate your additional comments and suggestions to improve the presentation of our figures and tables, and clarify the relevant parts of the text in revision. Thanks for the suggestions!

---

> > ### Comment · Reviewer_U13Y · 2025-04-02
> >
> > Thank you for your response. However, I feel that it did not fully address my concerns. To clarify, here are my thoughts:
> > 1. I fully understand that DP training is an important topic of ICML. However, could you please provide an example demonstrating whether any well-known models, such as GPT, Claude, Grok, DeepSeek, Qwen, or Llama, incorporate DP during their pre-training phase? I guess the main reason these companies do not use DP pre-training is that the model has bad performance over downstream tasks after DP pre-training. However, the authors do not show results over downstream tasks in the rebuttal.
> > 2. I am interested in building scaling laws primarily to train a large-scale model that achieves superior performance. However, this paper does not explain how to apply the scaling law to train such models effectively. I believe that targeting models in the 1B-2B parameter range is a reasonable expectation.
> > 3. Given that the NVIDIA A100 GPU comes with 40 GB of memory, it is feasible to fit a 1B or 2B model using data parallelism or Fully Sharded Data Parallelism (FSDP). I am not sure why they claim that "Scaling to larger models introduces significant engineering challenges, most notably handling model parallelism".
> > 4. Ultimately, the majority of renowned models have adopted a decode-only architecture. Given the current trends, I believe it will be impractical to use an encoder-decoder model in 2025. [7] observe that similar scaling between encoder-decoder and decoder-only architectures over Time-Series. It is not clear whether they have similar results over DP pre-training. Both [1] and [2] from Google DeepMin and OpenAI focus on using a decoder-only transformer model. Therefore, I think using Encoder-decoder Models is not a good choice.
> >
> > [7] Yao et al., Towards Neural Scaling Laws for Time Series Foundation Models
> >
> > [1] Hoffmann, Jordan, et al. "Training compute-optimal large language models." arXiv preprint arXiv:2203.15556 (2022).
> >
> > [2] Kaplan, Jared, et al. "Scaling laws for neural language models." arXiv preprint arXiv:2001.08361 (2020).

---

> > > ### Author Response · Authors · 2025-04-03
> > >
> > > Thank you for engaging with our response, clarifying your points, and agreeing to revisit your initial review based on this discussion. Please find our responses below.
> > >
> > > 1. We do not believe the well-known models you listed have incorporated DP in their pre-training phase. Despite this, we think our work is still well-motivated, for reasons separate from the ones provided in our initial response (which you may disagree with).  We hope that you will agree with the following statements:
> > >
> > >     * There are situations where it is useful to train models on in-domain sensitive user data, where DP must be applied. [3, 5, 6] The tasks where in-domain data is useful may be more narrow than the downstream tasks used to evaluate frontier models, and cross entropy is actually a very natural evaluation metric for some tasks (e.g., next word prediction for mobile keyboards [3]).
> > >
> > >     * Due to the high compute requirements common in DP training, it is important to understand compute/privacy/utility trade-offs, and ensure the compute budget is allocated in an intelligent manner to optimize utility. [2, 4]
> > >
> > >     * No prior work has systematically studied (2), and our work provides a rigorous and thorough study of these questions, and useful guidance for practitioners.
> > >
> > > If you think this provides a stronger motivation for our work, we will be happy to feature this more prominently in our introduction in revision.
> > >
> > > 2. Can you please clarify what you mean by “superior performance”, and what that is relative to?  Our motivation is related to but distinct from the motivation for non-private scaling laws that you are referencing. As mentioned above, we are interested in building scaling laws to allocate a fixed amount of compute to deliver the best possible DP model. This is a bit different from the reviewers goal of training the largest possible model. In fact, our findings show that in some settings, aiming for the largest possible model is detrimental to utility.
> > >
> > > 3. We agree your expectation of training 1B-2B parameter models is reasonable and would strengthen the paper – we have taken your advice into consideration and will revisit our experiments on BertGiga (1.5 B parameters).  However, as you must evaluate our submission in its current form, we hope you will take into consideration the following three points:
> > >     * BertMega is 77.8% the way to 1B in terms of parameter count, which is on the same order of magnitude for the purposes of this study.
> > >     * For this research, we had access to TPUv3 resources, which does feature 32 GB of RAM.  However, for a 1.5B parameter model represented with 4-byte floats, the model state is 6 GB.  Combined with the gradient (6 GB) and the optimizer state (12 GB), the activation memory, and other training overheads, we were not able to run DP-SGD with pure data parallelism.  We will note that without per-example gradient clipping, we were able to run this experiment, suggesting some memory overhead of DP (or at least the implementation we used).
> > >     * As we showed experimentally, there are many settings where it is clearly suboptimal to train a 1B+ model with DP.  For example, with a privacy budget of 4 a data budget (# contributing users) of 10M, the optimal model size is in the 10s of millions, even with an infinite amount of compute.
> > >
> > > 4. We agree that decoder-only models are preferable for autoregressive text generation and hence particularly suitable for chat bot applications. However, we do not agree with the statement that “it will be impractical to use encoder-decoder models in 2025”. The right model architecture depends on the task being solved; for example, encoder-only models like BERT are well-suited for tasks like natural language understanding, sentence classification, and question answering, while encoder-decoder architectures are well-suited for sequence-to-sequence tasks like machine translation and document summarization [8].
> > >
> > > [8] Qiu et al., Pre-trained Models for Natural Language Processing: A Survey

---

### Official Review · Reviewer_1Cfq · 2025-03-14

**Overall Recommendation:** 4

**Summary:**

The paper formulates the problem of identifying scaling laws for differentially private training of Bert models, as identifying the optimal training configurations (model size, batch size, noise-batch ratio, and iterations) given fixed data, compute, and privacy budget. Here clipping thresholds and stepsize are fixed as constants across all configurations. Their approach contains three steps.

1. For a fixed but reasonably large physical batch size, fit a function $L(M, T, \bar{\sigma})$ that estimates the training loss of an M-parameter model after T iterations with a noise-batch ratio of $\bar{\sigma}$, by repeating the training for different configurations and smoothening and interpolating the training results.
2. For other batch size $B$, assume that as long as $M, T, \bar{\sigma}$ are fixed, their training loss are equal.
3. For each fixed data, compute, and privacy budget, enumerate all possible training configurations (batch size, noise-batch ratio, and iterations), and identify the configuration that minimizes the estimated training loss $L(M, T, \bar{\sigma})$.

They then draw some insights from the behavior of the fitted scaling laws.
1. There is a small but consistent trend that with larger privacy budgets, one should train a larger model with a smaller batch size and for more iterations than one would train with a smaller privacy budget
2. Optimal model sizes under privacy are much smaller than predicted by non-private scaling laws.
3. Given a fixed privacy/data budget, there is an inflection point where increasing the compute budget provides little to no benefit.
4. The ratio of the number of training tokens to model size increases with computing budget, especially for smaller privacy budgets. This matches the flat token-to-model ratio as predicted by the prior work for non-private training (Hoffmann et al. (2022)).

**Claims And Evidence:**

In general, the paper is well-written with clear claims and supporting details. However, there are a few places that are harder to interpret due to missing details.
1. In Figure 1, is the noise-batch-ratio fixed across all settings? If so, why is it reasonable to fix noise-batch-ratio?
2. Section 4.5, how is the noise-batch ratio computed here? Is it corresponding to the optimal configurations predicted by $L(M, T, \bar{\sigma})$ given fixed $M$ and $T$? If so, why is it reasonable to fix $M$ and $T$?
> We analyze how the noise-batch ratio behaves as a function of privacy budget (as measured by ε), compute budget (as measured by B), and data budget (as measured by N).

**Essential References Not Discussed:**

Related works are discussed thoroughly. However, some sentences missed citations when referring to prior works. E.g., Line 327 says "becomes nearly flat as predicted by the prior work" without any references.

**Experimental Designs Or Analyses:**

The experiment results are comprehensive and interesting. Other than a few minor clarity issues as discussed in Claims And Evidence, I only have one doubt regarding what the authors call "improvements in the noise-batch ratio" in Section 4.5 and "diminishing returns in terms of noise-batch ratio" in Section 4.1. I'm a bit confused about whether we are interested in the increase/decrease of noise-batch-ratio as worsening/improvement of training, rather than the direct eval loss. Maybe it is reasonable under fixed choices of other factors, e.g., training iterations, batch size, and model size, it would be good if the authors could clarify these points.

**Methods And Evaluation Criteria:**

The method is interesting, and the only limitation that I see is the assumption that the training loss curve under different batch sizes is similar, as long as $M, T, \bar{\sigma}$ are fixed. The authors also discussed this in Appendix C.3. and show that this assumption may underestimate the benefit of smaller batch sizes when the noise-batch ratio is large.

Another minor point that lacks clarify, is why it is reasonable to fix the learning rate and clipping threshold across all settings, rather than, e.g., using a larger clipping threshold for larger models.

**Other Comments Or Suggestions:**

NA

**Other Strengths And Weaknesses:**

NA

**Questions For Authors:**

See Claims And Evidence and Experimental Designs Or Analyses.

**Relation To Broader Scientific Literature:**

The paper offers a nice study of scaling laws for differentially private training and discusses many connections/differences compared to the literature on scaling laws for standard training.

**Theoretical Claims:**

NA

---

> ### Author Rebuttal · Authors · 2025-04-01
>
> We thank the reviewer for the careful review and feedback, and we are glad you liked the paper.  Below we respond to the main questions / critiques:
>
> **[Missing details]** We will be sure to update our discussion around Figure 1 and Section 4.5 to clarify what is shown.
> * Regarding Figure 1, we do not use a constant noise-batch-ratio here; instead it is varied through the privacy budget, batch size, iterations, and data budget (as shown in the diagram in Figure 2).
> * In Section 4.5, we fix M and T and only consider how the noise batch ratio (sigma / B) changes with respect to the Privacy, Data, and Compute budget.
>
> **[Fixed batch size assumption]** Our modeling assumption that the training loss primarily depends on the batch size through the noise batch ratio (Figure 2) is a limitation we acknowledge in Appendix A.  However, this modeling assumption was necessary to make this work practically feasible, as adding yet another independent variable to consider greatly increases the number of experiments and accelerator hours needed to conduct this research.
>
> As the reviewer points out, our ablations in Appendix C.3 and C.4 do study the effect of this variable in isolation, although we believe it will be interesting to revisit this in future work to better understand the phenomenon we observed and under what conditions it manifests.
>
> **[Fixed clipping + learning rate]** We lean on findings from prior work to establish a reasonable default DP training setup.  In particular, [1] advocates for using a small clipping threshold where all (or nearly all) gradients are clipped.  We used a clipping threshold of 1, and checked that this resulted in > 90% of gradients being clipped.  We also used gradient normalization as [1] also suggests, since it decouples the clipping parameter from the learning rate. While more careful tuning of this parameter could help somewhat, our primary goal in this work was to focus on variables that relate directly to the compute budget.
>
> Regarding learning rate, we actually used three learning rates, which were chosen based on comprehensive ablations discussed in Appendix C.7 and Figure 12.
>
> [1] De et al., Unlocking High-Accuracy Differentially Private Image Classification through Scale

---

### Official Review · Reviewer_qUEy · 2025-03-17

**Overall Recommendation:** 4

**Summary:**

This paper explores the scaling laws applicable to the training of masked language models under differential privacy (DP) constraints. The authors establish that traditional scaling laws, which do not account for privacy considerations, are suboptimal when applied in DP settings. Key findings: optimal model size with DP is generally much smaller compared to non-private models.

**Claims And Evidence:**

The paper’s claims are supported by clear and methodical evidence, with experiments spanning model sizes, noise levels, and compute budgets. Limitations are transparently addressed, and the findings provide actionable insights for DP training.

I think it should be clearly stated that these findings are specific to MLM earlier than sec2, and the authors should make educated guesses about the transferability of the findings.

**Essential References Not Discussed:**

None

**Experimental Designs Or Analyses:**

The experimental design is methodologically generally rigorous in all the experiments for the tested settings (BERT models, masked LM, ≤778M parameters). However, critical assumptions—fixed batch sizes, reliance on (ϵ,δ)-DP, and limited architectural scope—constrain broader validity.

**Methods And Evaluation Criteria:**

Using BERT models (with varying sizes) and masked language modeling is appropriate, as BERT is a standard architecture for studying scaling laws. DP-Adam with per-example gradient clipping follows established DP-SGD practices. However, I’m concerned that results may not generalize to decoder-only models or tasks like autoregressive language modeling, which are common in modern LLMs. The fixed sequence length (512 tokens) and focus on pre-training (not fine-tuning) also limit direct applicability to real-world deployment scenarios.

Evaluation metrics like cross-entropy loss and compute savings are standard and meaningful, but the lack of diverse tasks/datasets and the reliance on only (ϵ,δ)-DP constrain generalizability.

**Other Comments Or Suggestions:**

* Algo 1 line 64: should that not be $=\bar{g}+...$ instead of $=g+$?
* left col line 84 which typically added. line 86 is
* line 121 left: different than
* line 718 is linear
The following recent refrence is not essential but might be useful in sec1 or sec5 as it gives a reasoned overview: Miranda+ 2024 https://arxiv.org/abs/2408.05212

**Other Strengths And Weaknesses:**

Strengths
* The paper is rich with practical insights, eg line 91 right, sec3.1, annex B2, annex D.
* The computational exploration is particularly broad and thorough, which gives the claims a solid grounding.

Weaknesses
* The caveat that the main findings are limited to BERT should be more prominent: neither title nor abstract address this.
* Citations in sec1 and sec 2 seem biased towards work by authors affiliated with Google without apparent reason. This should be corrected.
* Line 015 right: both citations are to blog posts; is there no better source?

**Questions For Authors:**

How do these findings transfer to causal language modeling?

**Relation To Broader Scientific Literature:**

The contribution is quite useful as it starts providing a guidance on scaling laws in this field. However, the impact is limited to the chosen setup and there are doubts on generalizability.

**Theoretical Claims:**

The claim that non-private scaling laws are suboptimal under DP is supported by empirical evidence shown in figures. Optimal model sizes are smaller under DP, it is empirically proven.

---

> ### Author Rebuttal · Authors · 2025-04-01
>
> We thank the reviewer for the careful review and feedback, and for recognizing the various strengths of this work.  Below we respond some of the questions / criticisms:
>
> **[Masked Language Modeling vs. Other Tasks]** The reviewer is correct to point out that our focus on BERT models raises questions about generality of the results with respect to other model architectures, like decoder-only models used in many modern GenAI applications.  We acknowledged this limitation in Appendix A.  We further provide evidence in Section 3.7 that despite the differences in training setups with prior (non-private) scaling laws work, we are able to reproduce the main finding of Hoffman et al., i.e., that data and model size should be increased in equal proportion with increasing compute.  This provides evidence that our results should translate to other settings.
>
> Our choice to study BERT models in this research was based on a few considerations:
> * Established pre-determined model sizes and configurations (https://github.com/google-research/bert)
> * Maturity of DP training support in distributed ML libraries (at the time this research started): We decided to use PaxML (https://github.com/google/paxml/) to conduct this research, since it supports large-scale training of transformer models coordinated across many accelerators, and supports per-example gradient clipping routines which are necessary to use DP-SGD that can run in distributed settings.  This open source library does not feature an easily-forkable implementation of a standard decoder-only model and thus would require a large effort to re-engineer purely from the available primitives.
>
> We believe that future research in this space should probably study (now) open-source decoder-only models like NanoGPT (https://github.com/karpathy/nanoGPT).  We believe that the findings we showed are still useful and informative in their current form, however.
>
> **[Pretraining vs. Finetuning]** The reviewer is correct to point out that with DP, finetuning has some advantages over pretraining, and would be interesting to study.  We acknowledged this limitation in Appendix A, and agree with the reviewer that this would be an interesting direction for future work.
>
> **[Evaluation task diversity]** This is a valid criticism, and something we will be sure to add to our Limitations section in revision, and highlight this as an important consideration for future work.
>
> **[Other weaknesses]** We have updated our references in the uploaded revision to include a more expansive set of citations. If the reviewer is aware of any relevant citations that were missed, we are happy to consider and add these as well. We have fixed the other minor issues identified during review, including the references on line 15 and the notational errors.  Thanks for the feedback!

---

> > ### Comment · Reviewer_qUEy · 2025-04-08
> >
> > I have read all reviews and rebuttal exchanges, including the interesting exchange between fellow Reviewer U13Y and Authors.
> >
> > I maintain my score, considering the trade-offs between the following arguments.
> >
> > * For the benefit of the paper, authors should definitely clarify important assumptions and goals including limitations: eg summarize motivation for decoder-only vs encoder-decoder models; practical goal of scaling laws in determining optimal model size ("engineering" knob) given data and compute budgets (often extrinsic constraints, not under the model training engineering team's influence); loss as proxy for downstream task performance. Certainly, I consider that mentioning limitation to decoder-only as little as in the current paper version is insufficient, opaque, and detrimental to the paper's resonance.
> > * The paper is particularly well-written; it can be improved locally; that would profile it for an oral for example.
> > * Conditional on agreeing with the work's motivations (which motivates making these strongly explicit), experiments are conclusive.

---

### Decision · Program_Chairs · 2025-05-01

**Decision:**

Accept (poster)

**Comment:**

This paper provides a careful empirical study of how differential privacy affects language model scaling laws. The authors demonstrate how scaling behaviors differ significantly from non-private regimes. Reviewers broadly commended the thoroughness of the experimental design and appreciated how the results could help practitioners allocate compute more effectively under privacy constraints, despite some reservations regarding real-world adoption of DP pre-training and a preference for decoder-only architectures in modern LLM use cases.

Nevertheless, the paper is clair and rigorous, and its relevance to DP gained strong support. The authors also addressed concerns in their rebuttal. Overall, this is a nice contribution.